# Target-Oriented Pretraining Data Selection via Neuron-Activated Graph

Zijun Wang [1 2]  Haoqin Tu [2]  Weidong Zhou [1]  Yiyang Zhou [3]  Xiaohuan Zhou [1]  Bingni Zhang [1]
Weiguo Feng [1]  Taifeng Wang [1]  Cihang Xie [2]  Fengze Liu [1]

## Abstract

Everyday tasks come with a target, and pretraining models around this target is what turns them into experts. In this paper, we study target-oriented language model (LM) pretraining by introducing *Neuron-Activated Graph Ranking* (NAG-based Ranking), a training-free and interpretable framework for target pretraining data selection. Rather than using black-box representations, our approach directly characterizes each target input by a sparse set of high-impact neurons in any off-the-shelf LLMs. Concretely, we quantify neuron impact and select the most influential neurons across layers into a compact *Neuron-Activated Graph* (NAG), and rank candidate data by NAG similarity to target examples. We conduct experiments across six benchmarks, where our NAG-based Ranking improves target-oriented pretraining by 4.9% on average over random sampling, and also outperforms state-of-the-art baselines by 5.3% accuracy on HellaSwag. It also remains effective under a more applicable multi-target setting, where our best setup surpasses two baselines by 1.1% and 4.1%, respectively. Furthermore, we provide a comprehensive analysis on *why* and *how* our NAG works, *e.g.*, deactivating NAG-selected neurons (only 0.12% of all) causes a 23.5% performance collapse, and restricting NAG to the final layer incurs a 4.1% average drop, indicating that NAG captures a sparse "functional backbone" for learning target features. We release the code at https://github.com/asillycat/NAG.

[1]ByteDance [2]UC Santa Cruz [3]UNC-Chapel Hill. Correspondence to: Zijun Wang <zwang745@ucsc.edu>, Fengze Liu <fengze.liu@bytedance.com>.

*Proceedings of the 43rd International Conference on Machine Learning*, Seoul, South Korea. PMLR 306, 2026. Copyright 2026 by the author(s).

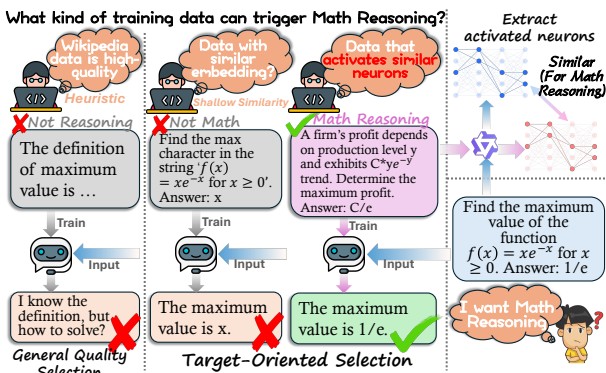

*Figure 1.* General quality-based data selection is often misaligned with specific downstream capabilities (*left*), while prior target-oriented methods rely on shallow similarity to target examples (*middle left*). Our NAG instead aligns pretraining data with target tasks by selecting inputs that activate similar neurons in the LLM, capturing the underlying capability required for the target (*middle right*), even across different domains (*e.g.*, economics *vs.* math).

## 1. Introduction

Large language models (LLMs) have become increasingly prevalent in everyday tasks, and people usually use these models with specific targets in mind. Selecting high-quality pretraining data is one of the most effective ways to improve model performance within a target domain, yielding great gains (Penedo et al., 2024; Mizrahi et al., 2025; Gunasekar et al., 2023; Sorscher et al., 2023). Despite its importance, what makes "high-quality" data remains surprisingly underdefined (Fig. 1). We argue that, in real-world settings, high-quality data should align with targeted scenarios that enable LLMs to acquire desired capabilities efficiently — education, medicine, or specific research domains — while ruling out other irrelevant factors that do not contribute to such capabilities (Mizrahi et al., 2025).

However, existing pipelines for data selection make this alignment ambiguous. Many rely on heuristic rules (Wenzek et al., 2019; Rae et al., 2022; Lee et al., 2022; Abbas et al., 2023) or implicit assumptions about "quality" (Sachdeva et al., 2024; Wettig et al., 2024; Penedo et al., 2024), leaving a noticeable gap between how data are chosen and the specific capabilities the model ultimately needs to develop.

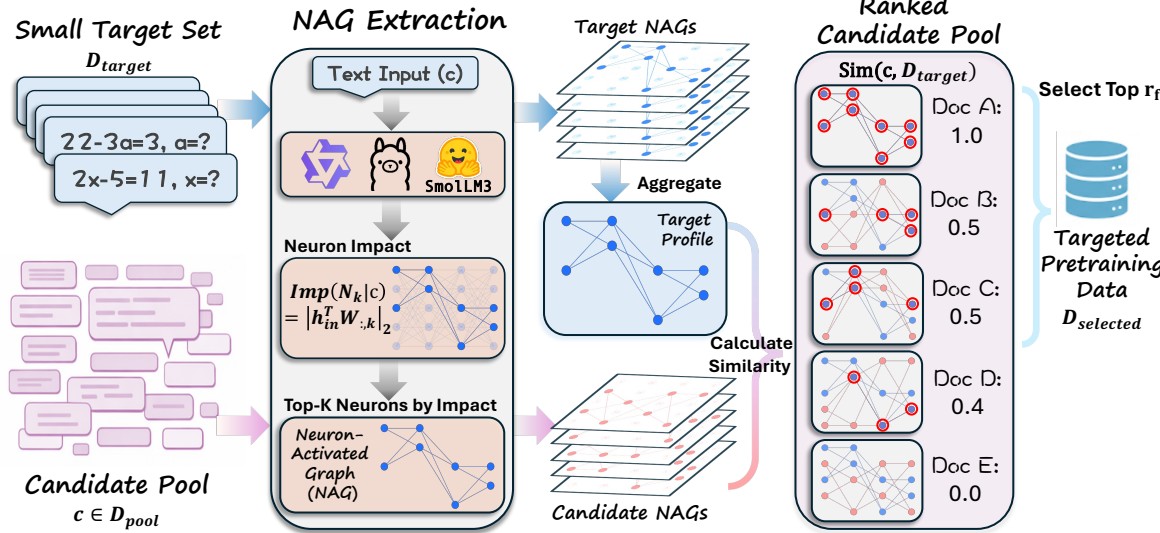

*Figure 2.* Overview of Neuron-Activated Graph (NAG) target-oriented data selection. Given a small set of target examples $D_{target}$, we first characterize each input by its neuron-level NAG features. For a given input, we quantify the impact of individual neurons and select top-$K$ of them per layer to construct a compact NAG. NAGs from the target examples are aggregated into a target neuron-activation profile. Each candidate sample $c \in D_{pool}$ is then mapped to its own NAG and ranked by its similarity to the target profile, *i.e.* $\text{Sim}(c, D_{target})$. Finally, the top-$r_f$ ranked samples are selected for LLM pretraining.

Prior efforts to align LLM pre-training data with explicit targets show that target-oriented pre-training can yield substantial compute multipliers and consistent gains across different scales (Mizrahi et al., 2025). However, most of these approaches achieve task alignment by distilling models' internal signals, such as embedding heuristics or performance-correlated losses, into auxiliary classifiers for scalability (Mizrahi et al., 2025; Thrush et al., 2025; SHUM et al., 2025; Miyoshi et al., 2025). This black-box distillation introduces an interpretability bottleneck: the learned signals are opaque and hard to diagnose or refine, thus constraining how effectively the signals can be leveraged for further performance gains.

To solve this, we introduce *Neuron-Activated Graph Ranking (NAG-based Ranking)*, a neuron-centric framework for target-oriented pretraining data selection. As illustrated in Fig. 2, our key idea is to characterize each text input by identifying which neurons in the LLM matter for processing it, rather than through black-box representations. Specifically, we first quantify neuron impact during inference in an off-the-shelf LLM (Sec. 2.1) and then organize the most influential neurons across layers into a compact *Neuron-Activated Graph (NAG)* (Sec. 2.2). For the data selection, samples are ranked by the similarity of their NAGs to the target examples, prioritizing training inputs that trigger similar neuron patterns with the target data. Notably, our NAG-based Ranking requires no additional training and relies solely on interpretable signals from any off-the-shelf LLM.

Empirical evaluations demonstrate our NAG-based Rank-ing consistently enhances target-oriented task performance across different settings and benchmarks. Compared to random sampling, NAG-based Ranking yields a substantial 4.9% average improvement in target-oriented pretraining, outperforming both strong classifiers focusing on general data quality (Penedo et al., 2024) and state-of-the-art methods on target-oriented data selection like BETR (Mizrahi et al., 2025). Moreover, our algorithm presents more general use in the multi-target scenario where baselines often degrade, with the best setup outperforming two baselines by 1.1% and 4.1%, respectively. We also prove the broadened application of NAG by adding its signals to another quality-based data selection method, which further boosts scores. Finally, the performance gains of NAG remain consistent across various backbone models from 4.7% to 5.0%, indicating the robust and model-agnostic nature of NAG.

Obtaining better performance is one aspect; explaining *how* and *why* NAG works makes our algorithm truly interpretable. We design dedicated experiments to probe from these two perspectives: (i) **Why NAG works:** we find that NAG works by identifying a sparse "functional backbone" of the LLM — deactivating just 0.12% of model neurons triggers a sharp 23.5% performance collapse (Sec. 4.1.1). These high-impact neurons can well represent different target information via clustering (Sec. 4.1.2) — NAG captures the true drivers of the desired targets. Finally, we show a high correlation between NAG-based rankings and the target learning utility (Sec. 4.1.3). (ii) **How NAG operates:** we spot the path on how NAG works, specifically, it maps a "computational trajectory" by aggregating signals across all

LLM layers. Our analysis shows that restricting NAG to the final layer results in a 4.1% average performance drop (Sec. 4.2.2). Furthermore, by twitching model components where NAG signals come from, we further identify that extracting signals from the internal projections (Sec. 4.2.1) at a very sparse neuron ratio (Sec. 4.2.3) is crucial for capturing effective task-specific signals.

## 2. Method

Prior works (Panigrahi et al., 2023; Zhao et al., 2024; 2025) suggest that model behavior is governed by a highly sparse subset of parameters, with different tasks relying on largely disjoint regions. As a result, inputs attributing to the same capability are expected to induce similar internal parameter usage patterns.

Based on this insight, we characterize each input by the subset of neurons that exert strong influence on the model's computation, and measure input relevance via the similarity of such neuron-level structures. We thus propose a neuron-centric framework for target-oriented data selection, which (1) quantifies neuron impact for each input (Sec. 2.1), (2) organizes the most influential neurons across layers into a compact *Neuron-Activated Graph (NAG)* (Sec. 2.2), and (3) ranks candidate samples by their NAG similarity to a small set of target examples (Sec. 2.3).

### 2.1. Neuron Impact

Following PLND (Zhao et al., 2024), we define a *neuron* as a column of a projection weight matrix in a Transformer-based language model, and focus on projection layers in both Attention (Q, K, V) and FFN (UP, DOWN) modules (Vaswani et al., 2017). Concretely, for a projection matrix $W \in \mathbb{R}^{d_{\text{in}} \times d_{\text{out}}}$, each column of $W$ is treated as an individual neuron, yielding $d_{\text{out}}$ neurons for that layer. For example, the FFN UP projection $W_{\text{up}} \in \mathbb{R}^{d_{\text{model}} \times d_{\text{internal}}}$ contains $d_{\text{internal}}$ neurons.

To quantify the contribution of a neuron $N_k$ in projection layer $\ell$, we measure the change induced by deactivating it. Since evaluating its effect on the final output is expensive, we adopt a local approximation and define neuron impact based on the corresponding projection layer output (we validate this local proxy against loss change in Sec. C).

Given input $h_{\text{in}} \in \mathbb{R}^{d_{\text{in}}}$, the layer output is $h_{\text{out}} = h_{\text{in}}^\top W$. Deactivating $N_k$ amounts to zeroing the $k$-th column of $W$, denoted as $W \backslash N_k$. We define the neuron impact as

$$\text{Imp}(N_k \mid h_{\text{in}}) = \left\| h_{\text{in}}^\top W - h_{\text{in}}^\top (W \backslash N_k) \right\|_2 = \left| h_{\text{in}}^\top W_{:,k} \right|,$$

where $W_{:,k}$ denotes the $k$-th column of $W$. This formulation shows that the neuron impact reduces to the magnitude of its column-wise contribution to the layer output.

In what follows, we refer to neurons with relatively large impact values as *activated* neurons for the given input.

### 2.2. Neuron-Activated Graph (NAG)

Building on the neuron impact defined in Sec. 2.1, we construct a structured, layer-wise representation of neuron impact patterns, termed the *Neuron-Activated Graph (NAG)*. We treat neurons with high impact scores as *activated* neurons for a given input, and use them as the basic units of the NAG. For each layer, we rank neurons by their impact scores and select a fixed number $K$ of high-impact neurons.

Consider a model with $L$ layers, where layer $\ell$ contains $d_\ell$ neurons. Given an input $c$, let $I_{\ell,k}(c)$ denote the impact of neuron $k$ in layer $\ell$. For each layer $\ell$, we select the indices of the top-$K$ neurons ranked by impact:

$$N_\ell^{(K)}(c) = \text{TopK}\left( \{I_{\ell,k}(c)\}_{k=1}^{d_\ell} \right) \subseteq \{1, \ldots, d_\ell\},$$

where $\text{TopK}(\cdot)$ returns the indices of the $K$ largest elements.

The *Neuron-Activated Graph (NAG)* of input $c$ is then defined as the collection of layer-wise neuron index sets:

$$\text{NAG}(c) = \left( N_1^{(K)}(c), N_2^{(K)}(c), \ldots, N_L^{(K)}(c) \right),$$

or equivalently as the set of layer–neuron index pairs

$$\text{NAG}(c) = \left\{ (\ell, k) \mid \ell \in \{1, \ldots, L\}, \ k \in N_\ell^{(K)}(c) \right\}.$$

### 2.3. Data Selection via NAG-Based Ranking

The Neuron-Activated Graph (NAG) provides a compact, neuron-level description of how an input is processed by the model during inference. Inputs with similar NAGs therefore exhibit similar neuron-level processing patterns within the model. We hypothesize that such structural similarity is indicative of shared task-relevant properties between inputs, which is further explored in Sec. 4.1.2.

Based on this hypothesis, we rank data samples according to their alignment with a target group, measured via NAG-based similarity.

**NAG-based similarity.** For two inputs $c$ and $c'$, we define their NAG-based similarity as

$$\text{Sim}(c, c') = \frac{2 \left| \text{NAG}(c) \cap \text{NAG}(c') \right|}{\left| \text{NAG}(c) \right| + \left| \text{NAG}(c') \right|}.$$

To compare an individual input against a group of inputs, we aggregate NAG statistics over the group. Given a dataset $\mathcal{D}$, for each layer–neuron index pair $(\ell, k)$ we compute the

*Table 1.* Results of NAG-based data selection under Single-Target and Multi-Target settings (Sec. 3.3). NAG is instantiated with different backbone models (*e.g.*, NAG$_{\text{Qwen3-1.7B}}$). For each benchmark, **bold** represents the best, and underlined represents the second; the best within the Single/Multi-Target settings is additionally highlighted with shade . Improvements are relative to Random and reported as subscripts (red for gains and blue for drops).

| Method | ARC-C | HellaSwag | TriviaQA | MMLU | XStoryCloze | XWinograd | Avg. |
|---|---|---|---|---|---|---|---|
| Random | 28.5% | 51.6% | 15.6% | 30.2% | 67.1% | 76.5% | 44.9% |
| FineWeb-Edu | 34.3%$_{+5.8\%}$ | 55.3%$_{+3.7\%}$ | 20.1%$_{+4.5\%}$ | **32.8%**$_{+2.6\%}$ | 65.9%$_{-1.2\%}$ | 76.2%$_{-0.3\%}$ | 47.4%$_{+2.5\%}$ |
| | | | Single-Target | | | | |
| BETR | 32.3%$_{+3.8\%}$ | 57.5%$_{+5.9\%}$ | 20.2%$_{+4.6\%}$ | 31.1%$_{+0.9\%}$ | **71.0%**$_{+3.9\%}$ | **80.7%**$_{+4.2\%}$ | 48.8%$_{+3.9\%}$ |
| NAG$_{\text{Qwen3-1.7B}}$ | 34.0%$_{+5.5\%}$ | **60.6%**$_{+9.0\%}$ | 22.3%$_{+6.7\%}$ | 32.2%$_{+2.0\%}$ | 70.0%$_{+2.9\%}$ | 80.1%$_{+3.6\%}$ | 49.8%$_{+4.9\%}$ |
| NAG$_{\text{Llama-3.2-3B}}$ | **35.0%**$_{+6.5\%}$ | 58.6%$_{+7.0\%}$ | 21.3%$_{+5.7\%}$ | 31.5%$_{+1.3\%}$ | 70.8%$_{+3.7\%}$ | 80.6%$_{+4.1\%}$ | 49.6%$_{+4.7\%}$ |
| NAG$_{\text{SmolLM3-3B}}$ | **35.0%**$_{+6.5\%}$ | 59.8%$_{+8.2\%}$ | **22.6%**$_{+7.0\%}$ | 31.2%$_{+1.0\%}$ | 70.5%$_{+3.4\%}$ | 80.6%$_{+4.1\%}$ | **49.9%**$_{+5.0\%}$ |
| | | | Multi-Target | | | | |
| BETR | 30.3%$_{+1.8\%}$ | 49.3%$_{-2.3\%}$ | 11.6%$_{-4.0\%}$ | 29.9%$_{-0.3\%}$ | 69.5%$_{+2.4\%}$ | 76.1%$_{-0.4\%}$ | 44.4%$_{-0.5\%}$ |
| NAG$_{\text{Qwen3-1.7B}}$ | 33.4%$_{+4.9\%}$ | 57.8%$_{+6.2\%}$ | 19.2%$_{+3.6\%}$ | 31.5%$_{+1.3\%}$ | 69.3%$_{+2.2\%}$ | 79.9%$_{+3.4\%}$ | 48.5%$_{+3.6\%}$ |
| NAG$_{\text{Llama-3.2-3B}}$ | 32.0%$_{+3.5\%}$ | 54.9%$_{+3.3\%}$ | 18.0%$_{+2.4\%}$ | 31.4%$_{+1.2\%}$ | 69.8%$_{+2.7\%}$ | 79.9%$_{+3.4\%}$ | 47.6%$_{+2.7\%}$ |
| NAG$_{\text{SmolLM3-3B}}$ | 31.8%$_{+3.3\%}$ | 55.2%$_{+3.6\%}$ | 19.9%$_{+4.3\%}$ | 30.6%$_{+0.4\%}$ | 69.2%$_{+2.1\%}$ | 80.2%$_{+3.7\%}$ | 47.8%$_{+2.9\%}$ |

frequency of corresponding neuron being activated:

$$w_{\ell,k}(\mathcal{D}) = \frac{1}{|\mathcal{D}|} \sum_{c' \in \mathcal{D}} \mathbf{1}[(\ell, k) \in \text{NAG}(c')],$$

where $\{w_{\ell,k}(\mathcal{D}) \mid \ell \in \{1, \dots, L\}, k \in \{1, \dots, d_\ell\}\}$ represents the NAG-based group profile of $\mathcal{D}$. The similarity between an input $c$ and the group $\mathcal{D}$ is then defined as

$$\text{Sim}(c, \mathcal{D}) = \frac{1}{L} \sum_{\ell=1}^{L} \frac{\sum_{k \in N_\ell^{(K)}(c)} w_{\ell,k}(\mathcal{D})}{\sum_{k=1}^{d_\ell} w_{\ell,k}(\mathcal{D})}.$$

Under our setting where each sample selects exactly $K$ neurons per layer, this group similarity is equivalent to the average of pairwise similarities $\frac{1}{|\mathcal{D}|} \sum_{c' \in \mathcal{D}} \text{Sim}(c, c')$; the frequency form is a more efficient computation that avoids enumerating all pairs. See Sec. D for the full derivation.

**Target-oriented ranking and selection.** Let $\mathcal{D}_{\text{target}}$ denote a small set of target examples characterizing the desired task or domain. We construct a target NAG-based profile by aggregating NAG statistics over $\mathcal{D}_{\text{target}}$. Each candidate sample $c \in \mathcal{D}_{\text{pool}}$ is then assigned a similarity score

$$s(c) = \text{Sim}(c, \mathcal{D}_{\text{target}}).$$

We then rank all samples in $\mathcal{D}_{\text{pool}}$ in descending order of $s(c)$ and select the top fraction with a predefined ratio $r_f \in (0, 1]$:

$$\mathcal{D}_{\text{selected}} = \text{TopRatio}_{r_f}\big(\mathcal{D}_{\text{pool}}, s(\cdot)\big).$$

This procedure prioritizes samples whose neuron-level processing structures best align with the target group, enabling efficient and targeted pretraining data selection.

# 3. Experiments

## 3.1. Pretraining Source Data

We use RefinedWeb (Penedo et al., 2023), a high-quality, web-only English pretraining corpus containing approximately 600B tokens. We uniformly downsample the corpus to 150B tokens to construct a source data pool. All subsequent experiments perform *document-level* data selection from this pool. Specifically, each method ranks data samples by its own criterion and selects the top subset whose total token count reaches 30B tokens (*i.e.*, 20% of the pool), which is then used for pretraining.

## 3.2. Benchmarks

We evaluate on six widely used benchmarks, spanning a broad range of reasoning skills, including multiple-choice reasoning (ARC-Challenge (Clark et al., 2018), HellaSwag (Zellers et al., 2019), MMLU (Hendrycks et al., 2021)), factual question answering (TriviaQA (Joshi et al., 2017)), narrative understanding (XStoryCloze (Lin et al., 2022)), and commonsense reasoning (XWinograd (Tikhonov & Ryabinin, 2021)). All evaluations are conducted using the `lm-eval-harness` framework (Gao et al., 2024), following the official evaluation splits and default prompting templates. These benchmarks largely overlap with those used in recent data selection studies (Liu et al., 2025a; Hua et al., 2025; Sachdeva et al., 2024; Mizrahi et al., 2025). See Sec. A.2 for benchmark details.

## 3.3. Setup

**Training.** We use transformer architecture (Vaswani et al., 2017), SwiGLU (Shazeer, 2020) activation function and RoPE embeddings (Su et al., 2024). We use a tokenizer

*Table 2.* Results of integrating NAG-based ranking with the FineWeb-Edu quality signals under the Single-Target setting. Improvements are relative to FineWeb-Edu and shown in red. The best are shown in shade .

| Method | ARC-C | HellaSwag | TriviaQA | MMLU | XStoryCloze | XWinograd | Avg. |
|---|---|---|---|---|---|---|---|
| FineWeb-Edu | 34.3% | 55.3% | 20.1% | 32.8% | 65.9% | 76.2% | 47.4% |
| + NAG$_{Qwen3-1.7B}$ | 35.3%$_{+1.0\%}$ | 57.7%$_{+2.4\%}$ | 21.7%$_{+1.6\%}$ | 32.5%$_{-0.3\%}$ | 67.2%$_{+1.3\%}$ | 79.2%$_{+3.0\%}$ | 48.9%$_{+1.5\%}$ |
| + NAG$_{Llama-3.2-3B}$ | 35.2%$_{+0.9\%}$ | 57.4%$_{+2.1\%}$ | 21.7%$_{+1.6\%}$ | 32.7%$_{-0.1\%}$ | 68.2%$_{+2.3\%}$ | 78.6%$_{+2.4\%}$ | 49.0%$_{+1.6\%}$ |
| + NAG$_{SmolLM3-3B}$ | 35.7%$_{+1.4\%}$ | 58.1%$_{+2.8\%}$ | 22.7%$_{+2.6\%}$ | 33.1%$_{+0.3\%}$ | 69.0%$_{+3.1\%}$ | 78.9%$_{+2.7\%}$ | 49.6%$_{+2.2\%}$ |

with 250k vocabulary and a model structure using 1.2B parameters. See Sec. A.1 for details about model structure, learning rate and optimizer. All models are trained from scratch with identical architectures, optimization settings, and a fixed budget of 30B training tokens. The only difference across experiments lies in the data selection strategy.

**NAG Configuration.** Unless otherwise specified, we use the UP projection layers for NAG construction (see Sec. 4.2.1). For data selection, we fix the filtering rate to $r_f = 20\%$. We use a width ratio of $r_k = K/d_\ell = 0.3\%$ for all layers (see Sec. 4.2.3), and report the corresponding layer-wise $K$ values in Tab. 8. We evaluate backbone dependence by constructing NAG with Qwen3-1.7B-Base (Yang et al., 2025), Llama-3.2-3B (Grattafiori et al., 2024), and SmolLM3-3B (Bakouch et al., 2025).

**Targeting Setting.** We evaluate our method under two targeting settings to assess both specialization and generalization. To ensure evaluation integrity, target examples are drawn exclusively from the *training splits* of the benchmarks, and we perform careful decontamination against all benchmark test sets (see Sec. A.3 for details).

- **Single-Target.** We consider a target-specific data selection scenario, where each experiment focuses on a single benchmark. The benchmark is treated as the target center, and we select the top-ranked samples from the source pool according to a fixed filtering rate $r_f = 20\%$.

- **Multi-Target.** To adapt to real-world scenarios, we further evaluate robustness under mixed objectives by using multiple targets simultaneously. All six benchmarks are treated as target centers. For each target, we independently select samples using an equal share of the selection budget (*i.e.*, $r_f/6$), then directly mix resulting subsets.

### 3.4. Baselines

**Random** samples 20% of the 150B-token data pool introduced in Sec. 3.1 randomly as the pretraining data, which is a widely-used baseline in existing literature (Mizrahi et al., 2025; Wettig et al., 2024; Sachdeva et al., 2024).

**FineWeb-Edu Classifier** (Penedo et al., 2024) ranks pretraining samples according to their estimated educational

value using a learned classifier.

**BETR** (Mizrahi et al., 2025) proposes a task-matching data selection method that ranks pretraining samples based on their embedding similarity to a set of target examples.

We follow the original setups of baseline papers and use the same targeting settings and filtering rate as in our method to ensure a fair comparison.

### 3.5. Main Results

Tab. 1 summarizes the main results under both the single-target and multi-target settings, with performance of random sampling, two strong data selection baselines, and our method across three LLM backbones.

**NAG-based Ranking Consistently Enhances Target Performance.** Compared with random selection, our method yields an average improvement of 4.9% across target benchmarks. When comparing with the strong FineWeb-Edu baseline that focuses on task-agnostic quality heuristics, our selected data achieves better performance overall (+2.4%). Notably, the gains are most pronounced on benchmarks that are underrepresented by general quality heuristics, such as HellaSwag (+4.4%), XStoryCloze (+4.5%), and XWinograd (+4.2%), indicating that NAG captures task-relevant signals that complement general-quality-focused selection.

On the other hand, NAG-based Ranking outperforms the target-oriented method BETR by 1% on average, especially on ARC-C (+2.4%) and HellaSwag (+2.2%). We hypothesize this stems from the nature of the similarity signals: while BETR relies on last-layer embeddings — which often conflate semantic and stylistic surface features (Lyu et al., 2023; Skean et al., 2025) — NAG aggregates neuron-level signals across all layers (Sec. 4.2.2). This allows our method to capture deeper and shared neuron patterns in LLMs that cover a wider range of task representations (Sec. 4.1.2).

We also note that these target-specific enhancements remain consistent across various backbone models used for data selection (+4.7%-5.0%), indicating the effectiveness of NAG does not depend on a specific model architecture.

**NAG-based Ranking Achieves Multi-Target Gains with Simple Data Mixture.** The targets of real-world applica-

*Table 3.* Targeted neuron deactivation on Qwen3-1.7B-Base. We deactivate only 0.12% of all neurons, selected either randomly or by NAG. Performance drops relative to the original model are shown in blue. Severe drops under NAG deactivation indicate that NAG identifies a sparse set of critical neurons.

| Method | ARC-C | HellaSwag | TriviaQA | MMLU | XStoryCloze | XWinograd | Avg. |
|---|---|---|---|---|---|---|---|
| Qwen3-1.7B-Base | 55.7% | 66.9% | 36.3% | 45.9% | 72.4% | 86.5% | 60.6% |
| Deactivate 20 neurons per layer (0.12%) | | | | | | | |
| Deactivate Random | 55.5%$_{-0.2\%}$ | 66.8%$_{-0.1\%}$ | 35.8%$_{-0.5\%}$ | 45.8%$_{-0.1\%}$ | 72.4%$_{-0.0\%}$ | 85.9%$_{-0.6\%}$ | 60.4%$_{-0.2\%}$ |
| Deactivate NAG | 30.4%$_{-25.3\%}$ | 45.6%$_{-21.3\%}$ | 0.3%$_{-36.0\%}$ | 29.1%$_{-16.8\%}$ | 56.9%$_{-15.5\%}$ | 60.6%$_{-25.9\%}$ | 37.1%$_{-23.5\%}$ |

tions might be multiple at a time. To evaluate the generalizability of our method under such settings, we adopt a simple multi-target selection strategy: subsets are independently selected for each target and directly merged without re-weighting or de-duplication. Under this naive mixture, BETR exhibits a substantial performance drop compared to the single-target setting (-4.4%), suggesting that direct mixture poses a challenging scenario and can serve as a lower bound for the multi-target setting. In contrast, our NAG-based Ranking consistently surpasses random selection (+3.1%) and strong FineWeb-Edu (+0.6%) under mixed targets. When using Qwen3-1.7B-Base as the backbone model, it achieves the best performance, with an average gain of 3.6% over random selection. These observations provide preliminary evidence in applying our approach to more generalizable multi-objective scenarios in more complex environments. Furthermore, more advanced data mixture strategies (*e.g.*, RegMix (Liu et al., 2025b) and QuaDMix (Liu et al., 2025a)) may further improve NAG under the multi-target setting, which we leave for future exploration.

### 3.6. Combining NAG with Existing Quality Signals

Beyond serving as a standalone selection signal, we explore whether NAG can support broader applications by complementing existing quality-based data selection methods. Specifically, we evaluate a joint ranking that combines NAG with the FineWeb-Edu classifier.

As shown in Tab. 2, the integrated ranking consistently outperforms FineWeb-Edu classifier alone, yielding an average improvement of 1.8% across all benchmarks. Notably, on the hard ARC-C task, the combined approach exceeds both the FineWeb-Edu-only and NAG-only baselines (*e.g.*, 35.4%>34.7%>34.3%), suggesting an additive effect between two signals. These results demonstrate that NAG captures information complementary to general-quality scores and can be seamlessly integrated into existing data pipelines to achieve more robust and effective selection.

### 4. Analysis

In this section, we theoretically analyze the interpretability of NAG, focusing on *why* and *how* it works.

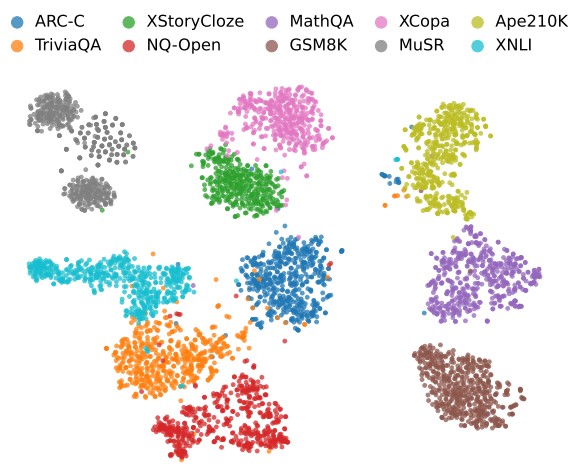

*Figure 3.* Task-level clustering of data instances based on NAG representations. The resulting clusters align closely with task identities, indicating NAG encodes task-discriminative representations.

### 4.1. Why NAG Works

We argue that NAG works because it (i) isolates critical neurons, (ii) organizes them into task-discriminative representations, and (iii) induces ranks that align sharply with downstream utility.

#### 4.1.1. NAG CAPTURES CRITICAL NEURONS

NAG characterizes each input using a sparse set of high-impact neurons in LLMs. To justify this design choice, we evaluate whether the neurons selected by NAG are indeed crucial to impact the model's final performance by selectively deactivating them (see Sec. B.1 for details). Tab. 3 demonstrates that deactivating NAG-selected neurons, though comprising only 0.12% of the total model neurons, induces a sharp 23.5% performance drop (from 60.6% to 37.1%), whereas deactivating an equivalent number of random neurons has a negligible effect. This contrast confirms that NAG isolates a highly sparse set of neurons with a higher-level of importance, justifying our focus on high-impact neurons.

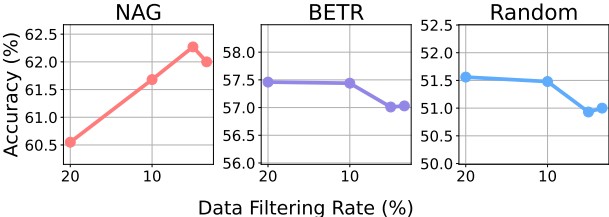

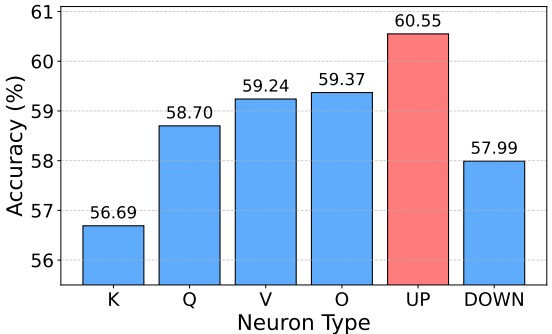

*Figure 4.* Performance under varying filtering rates $r_f$ for data selected by different ranking methods. Results are reported in the Single-Target setting with HellaSwag as the target; NAG is constructed from Qwen3-1.7B-Base using the default configuration. NAG consistently improves performance as lower-ranked samples are removed, indicating strong alignment between its induced ranking and downstream task utility.

*Figure 5.* Effect of neuron type on NAG construction. The results are reported on HellaSwag.

### 4.1.2. NAG IS A TASK-DISCRIMINATIVE REPRESENTATION

To give intuitive reasons on why NAG helps pretraining data focus on the given target, we visualize its representations at the task level. Specifically, we perform a dataset-level clustering by sampling 500 instances from ten diverse datasets (Sec. E) and computing pairwise NAG-based distances $d(c, c') = 1 - \text{Sim}(c, c')$. The resulting t-SNE visualization (Fig. 3) reveals clear, distinct clusters aligned with task identities. Also note that the relative positioning of clusters reflects task relevance, *e.g.*, MathQA and GSM8K, two benchmarks requiring mathematical reasoning, form two closer clusters, yet remain well-separated from linguistic tasks like XNLI. This demonstrates that NAG well captures task-relevant features, leading to reliable and effective target-oriented data selection.

### 4.1.3. NAG-BASED RANKS ALIGN WITH DOWNSTREAM TASK UTILITY

To justify that NAG actually provides meaningful data ranking for the follow-up selection instead of just a lucky guess, we design an experiment to assess the task performance with varied filtering rates $r_f$ of NAG ranking while keeping the source data fixed. As illustrated in Fig. 4, random selection leads to a performance drop (from 51.6% to 50.9%) as the filtering rate $r_f$ decreases, which might be attributed to a loss of data diversity. Similarly, BETR exhibits a degradation (-0.5%), suggesting its represented data features are weakly correlated to target data utility. In contrast, our NAG demonstrates a sharp performance increase under more aggressive filtering, peaking at $r_f = 5\%$ with 1.8% accuracy boost even with the highest base score at $r_f = 20\%$ (*e.g.*, 60.5% to 62.3%). This validates that NAG induces a utility-aligned ranking, where progressively removing lower-ranked samples improves downstream performance — NAG effectively ranks data by the target task utility.

## 4.2. How NAG Operates

In finding the reasons behind *how*, we analyze how design choices in which neuron type, which layer in the LLM, and which level of sparsity of neurons favor the use of NAG.

### 4.2.1. NAG PERFORMS BEST WITH FFN_UP NEURONS

We first investigate the impact of different neuron types on NAG construction by fixing $r_f = 20\%$ and $K = 20$ on Qwen3-1.7B-Base. Our comparison across various projection layers (Fig. 5) reveals that up_proj neurons achieve the highest performance at 60.6%, while down_proj and k_proj yield lower scores of 58.0% and 56.7%, respectively. We hypothesize this performance gap arises because expansion layers like up_proj operate in a higher-dimensional latent space that better isolates task-specific signals, whereas projection layers closer to the residual stream (down_proj, k_proj) tend to capture more compressed information. These results suggest selecting neurons from up_proj is most effective for identifying high-utility data.

### 4.2.2. NAG AGGREGATES TASK-RELEVANT SIGNALS ACROSS LAYERS

While many data selection methods rely on final-layer representations only (*e.g.*, last-layer embeddings (Mizrahi et al., 2025) or logits (SHUM et al., 2025; Thrush et al., 2025)), we investigate whether the power of NAG can be fully unleashed by leveraging more LLM layers. We compare the standard multi-layer NAG with a variant restricted to the final layer, with results summarized in Tab. 4. Our observation suggests that restricting NAG to the final layer leads to a substantial average performance drop of 4.1%. Notably, on challenging benchmarks like TriviaQA and MMLU, where the average accuracy of random selection is only 22.9%, the final-layer variant even underperforms random selection by 0.4%. These results demonstrate that task-relevant signals are distributed across the entire pack of model layers, justifying the design of the multi-layer deployment of NAG.

*Table 4.* Comparison between All-Layer and Last-Layer NAG constructed from Qwen3-1.7B-Base under Single-Target setting. Performance differences are shown in blue. Using only last-layer neurons leads to consistent performance degradation across all benchmarks, indicating that task-relevant signals are distributed across layers.

| Method | ARC-C | HellaSwag | TriviaQA | MMLU | XStoryCloze | XWinograd | Avg. |
|---|---|---|---|---|---|---|---|
| NAG$_{\text{All Layer}}$ | 34.0% | 60.6% | 22.3% | 32.2% | 70.0% | 80.1% | 49.8% |
| NAG$_{\text{Last Layer}}$ | 30.5%$_{-3.5\%}$ | 55.2%$_{-5.4\%}$ | 15.1%$_{-7.2\%}$ | 29.9%$_{-2.3\%}$ | 67.8%$_{-2.2\%}$ | 75.5%$_{-4.6\%}$ | 45.7%$_{-4.1\%}$ |

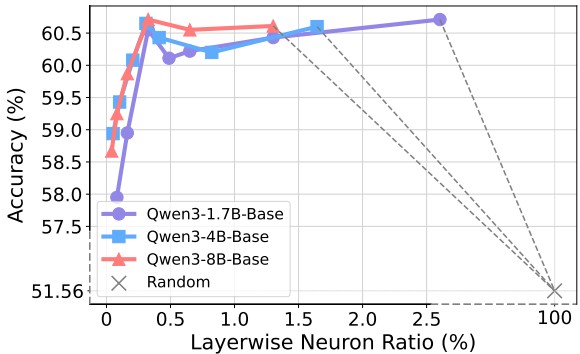

*Figure 6.* Effect of neuron sparsity (layerwise neuron ratio $r_k$) on NAG construction. NAGs are extracted from the Qwen3-Base family (1.7B, 4B, and 8B). Performance consistently peaks at $r_k = 0.3\%$ across model scales. As $r_k \to 1$, the NAG-based ranking theoretically collapses toward random selection.

### 4.2.3. NAG CONCENTRATES INTO A SPARSE SET OF NEURONS

While NAG has been shown to be effective, we ask: how sparse do the selected neurons need to be to achieve strong performance? We parameterize the sparsity using a layerwise neuron ratio $r_k$, where the number of neurons per layer $K = r_k \times d_\ell$, and evaluate NAGs extracted from the Qwen3-Base family (1.7B, 4B, and 8B) on the HellaSwag target. As shown in Fig. 6, performance increases rapidly as $r_k$ grows and reaches its maximum at $r_k \approx 0.3\%$ across different model scales. Further increasing $r_k$ yields only little or reverse gains; for example, on the 1.7B model, performance at $r_k = 2.1\%$ ($7\times$ neurons) is comparable to that at $r_k = 0.3\%$. This confirms that the most competent task-relevant signals are concentrated within a sparse set of high-impact neurons.

Moreover, we observe that NAG performance favors larger LLMs — under a fixed $r_k$, larger models consistently yield better performance. This suggests that increased model capacity offers more distinctive neuron representations, thus facilitating more useful task-oriented signals under NAG.

## 5. Related Works

**General-Quality Pretraining Data Selection.** A large body of prior work focuses on selecting *generally high-quality* pretraining data, without explicit consideration of downstream targets. One dominant paradigm relies on super-

vised or weakly supervised classifiers trained to distinguish "high-quality" text from noise, as exemplified by FineWeb-Edu (Penedo et al., 2024) and DCLM (Li et al., 2025). These approaches inherently depend on curated labels, which can introduce biases and entangle the notion of quality with the particular data sources or annotation heuristics used. Another line of work adopts heuristics such as perplexity-based filtering, language identification, or deduplication (Wenzek et al., 2019; Rae et al., 2022; Lee et al., 2022; Abbas et al., 2023). These methods similarly rely on coarse proxies of quality that are agnostic to the specific capabilities a model is expected to acquire. More recent efforts propose learning scalar quality scores or preference models (Sachdeva et al., 2024; Wettig et al., 2024), but these signals are still derived from model outputs or losses, reflecting only shallow, final-layer behavior. As a result, existing general-quality data selection methods largely overlook richer internal computation signals within the model, leaving a gap between data filtering criteria and the underlying mechanisms that govern capability learning.

**Target-Oriented Pretraining Data Selection.** Recent work has explored aligning pretraining data with specific downstream tasks, showing that task-aware data selection can substantially improve training efficiency and downstream performance. A common paradigm in this line of work is to estimate task relevance via proxy signals. BETR (Mizrahi et al., 2025) measures similarity between source data and target examples in a learned embedding space, constructs pseudo-labels based on similarity-ranking, and trains a lightweight classifier. SHUM et al. (2025) and Thrush et al. (2025) define proxy signals that are correlated with downstream benchmark performance—such as LM loss or perplexity—and distill these signals into lightweight classifiers to estimate data utility for pretraining data selection. DAIG (Miyoshi et al., 2025) follows a related proxy-based approach by training an auxiliary model on target data and using its predictions to score source data.

Overall, while differing in their specific proxy signals, these approaches infer task alignment *indirectly* through compressed black-box representations. In contrast, our approach derives task alignment directly from neuron-level computation in an off-the-shelf LLM, yielding an interpretable signal for task-oriented data selection.

# 6. Conclusion

In this work, we proposed NAG-based Ranking, a neuron-centric method for target-oriented pretraining data selection. Our approach represents each input with a Neuron-Activated Graph and ranks data by neuron-level similarity to target examples. It requires no additional training and relies only on interpretable signals from off-the-shelf LLMs. Experiments across benchmarks, target settings, and backbone models show consistent gains over random sampling and strong baselines. Beyond empirical improvements, our analyses reveal why NAG works: it isolates a sparse "functional backbone" of high-impact neurons and captures task-discriminative signals distributed across layers. We hope that this work encourages further exploration of neuron-level interpretability for data selection, and more broadly, for understanding and steering the capabilities learned during large-scale pretraining.

**Limitations and future work.** Our main experiments train a 1.2B model on 30B tokens from RefinedWeb; extending to larger models and more diverse corpora (*e.g.*, multilingual or domain-specific data) is left for future work, though our preliminary 7B results on HellaSwag (Sec. I) are encouraging. The multi-target setting uses a simple equal-budget mixture as a lower-bound scenario, and more advanced mixture strategies (*e.g.*, RegMix, QuaDMix) could further improve multi-target performance.

# Impact Statement

This paper presents work whose goal is to advance the field of machine learning. There are many potential societal consequences of our work, none of which we feel must be specifically highlighted here.

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

# A. Experimental setup details

## A.1. Training

The model we used has 1.2B parameters, its structure is illustrated in Tab. 5. We train all the model with 2048 as the max sequence length, we use a cosine decay schedular and the initial learning rate lr $= 5 \times 10^{-4}$, the end learning rate is lr $= 5 \times 10^{-5}$, the warm up ratio is set 0.5%. We use AdamW optimizer with $\beta_1 = 0.9$, $\beta_2 = 0.95$, weight decay$= 0.1$.

*Table 5.* Structure of model used in Sec. 3

| | |
|---|---|
| **Hidden dim.** ($d_{model}$) | 2048 |
| **MLP dim.** ($d_{internal}$) | 5440 |
| **Layers (L)** | 24 |
| **Heads** | 16 |

## A.2. Benchmark details

We evaluate model performance on six widely used reasoning and commonsense benchmarks, with detailed evaluation settings summarized in Tab. 6. All evaluations are conducted using the `lm-eval-harness` framework (Gao et al., 2024), following the official evaluation splits and default prompting templates.

For each benchmark, we adopt standard few-shot configurations that are commonly used in prior work to ensure fair and reproducible comparisons. Specifically, the number of in-context examples (shots) varies across benchmarks to reflect their established evaluation protocols, as reported in Tab. 6. When applicable, we report normalized accuracy to account for answer choice biases in multiple-choice settings; otherwise, we use task-specific metrics such as exact match or standard accuracy.

The selected benchmarks cover a diverse set of reasoning skills, including multiple-choice reasoning (ARC-Challenge, HellaSwag, MMLU), factual question answering (TriviaQA), narrative understanding (XStoryCloze), and coreference-based commonsense reasoning (XWinograd). Together, they provide a comprehensive and representative evaluation of both reasoning and knowledge-intensive capabilities, and largely overlap with benchmarks used in recent data selection studies.

*Table 6.* Benchmark details and evaluation settings. All results are obtained using `lm-eval-harness` (Gao et al., 2024). We report normalized accuracy when applicable and follow standard few-shot configurations used in prior work.

| Benchmark | Task Type | Shots | Test Size | Metric | Description |
|---|---|---|---|---|---|
| ARC-Challenge (Clark et al., 2018) | MC Reasoning | 25 | 1,172 | Acc$_{norm}$ | Grade-school science questions requiring multi-step reasoning and commonsense knowledge. |
| HellaSwag (Zellers et al., 2019) | MC Commonsense | 10 | 10,042 | Acc$_{norm}$ | Select the most plausible continuation of a short narrative from adversarial options. |
| TriviaQA (Joshi et al., 2017) | Factual QA | 5 | 17,944 | Exact Match | Open-domain factual question answering across diverse knowledge domains. |
| MMLU (Hendrycks et al., 2021) | MC Knowledge | 5 | 14,042 | Acc$_{norm}$ | Multi-domain academic reasoning benchmark covering STEM, humanities, and professional subjects. |
| XStoryCloze (Lin et al., 2022) | Narrative Understanding | 0 | 1,511 | Accuracy | Choose the coherent ending of a short story based on temporal and causal consistency. |
| XWinograd (Tikhonov & Ryabinin, 2021) | Coreference Reasoning | 5 | 2,325 | Accuracy | Pronoun resolution requiring semantic and contextual commonsense reasoning. |

### A.3. Target Set Construction and Decontamination

For each benchmark, we construct the target set $\mathcal{D}_{\text{target}}$ used for NAG extraction by sampling from the corresponding training or validation split. The details are summarized in Tab. 7.

*Table 7.* Target set details per benchmark.

| Benchmark | Size | Source split |
|---|---|---|
| ARC-C | 1,119 | full train split |
| HellaSwag | 10,000 | randomly sampled from train split |
| TriviaQA | 10,000 | randomly sampled from train split |
| MMLU | 1,531 | full validation split |
| XStoryCloze | 360 | full English train split |
| XWinograd | 2,124 | full non-English split |

**Decontamination.** The target sets used for NAG extraction are drawn exclusively from train/validation splits, which are by construction independent from the test splits used for evaluation. We additionally perform a 13-gram overlap decontamination check to verify that the target samples do not overlap with any benchmark test instances.

### A.4. NAG Width

Tab. 8 reports the effective NAG widths under different backbone models used in Tab. 1. When $r_k = 0.3\%$ (Sec. 4.2.3), the NAG width $K \approx r_k \times d_\ell$, where $d_\ell = d_{\text{internal}}$ in UP projection layer.

## B. Deactivation

This section provides full experimental details for the neuron deactivation analysis summarized in Sec. 4.1.1. And the full results are shown in Tab. 9.

### B.1. Experimental Setup

We conduct neuron deactivation experiments using NAGs constructed from a fixed-width setting with $K = 20$ neurons per layer. Unless otherwise specified, NAGs are extracted from UP neurons and then statistically grouped within 10k randomly sampled inputs. Neuron importance is quantified using the impact scores defined in Sec. 2, and neurons are deactivated by zeroing out their activations during inference.

### B.2. Coarse-Grained Deactivation: NAG vs. Random

We first compare the effect of deactivating NAG-selected neurons against a random baseline. For each layer, we deactivate the top-20 neurons selected by NAG, corresponding to approximately 0.12% of all neurons in the model. As a control, we deactivate the same number of neurons sampled uniformly at random per layer.

As shown in Tab. 9, deactivating NAG-selected neurons leads to a substantial average performance drop of 23.5% across tasks. In contrast, deactivating an equal number of randomly selected neurons results in negligible performance degradation. This demonstrates that neurons selected by NAG are both sparse and functionally critical.

*Table 8.* Effective NAG width under different backbone models used in Tab. 1. For each model, we report the MLP dimension (`up_proj`) $d_{\text{internal}}$ and the corresponding NAG width $K$, where $K \approx r_k \times d_{\text{internal}}$ with $r_k = 0.3\%$ (Sec. 4.2.3).

| Model | $d_{internal}$ | $K$ |
|---|---|---|
| Qwen3-1.7B-Base | 6144 | 20 |
| Llama-3.2-3B | 8192 | 20 |
| SmolLM3-3B | 11008 | 30 |

*Table 9.* Targeted neuron deactivation on Qwen3-1.7B-Base. We deactivate only 0.12% of all neurons, selected either randomly or by NAG. To further analyze which neurons within an NAG contribute most to performance, we additionally deactivate only 0.006% of all neurons using different selection criteria: random selection, consistently highly activated neurons (High-Mean), and neurons that induce large impact differences between target examples and random inputs (High-$\Delta$). Performance drops relative to the original model are shown in blue.

| Method | ARC-C | HellaSwag | TriviaQA | MMLU | XStoryCloze | XWinograd | Avg. |
|---|---|---|---|---|---|---|---|
| Qwen3-1.7B-Base | 55.7% | 66.9% | 36.3% | 45.9% | 72.4% | 86.5% | 60.6% |
| | | | Deactivate 20 neurons per layer (0.12%) | | | | |
| Deactivate Random | $55.5\%_{-0.2\%}$ | $66.8\%_{-0.1\%}$ | $35.8\%_{-0.5\%}$ | $45.8\%_{-0.1\%}$ | $72.4\%_{-0.0\%}$ | $85.9\%_{-0.6\%}$ | $60.4\%_{-0.2\%}$ |
| Deactivate NAG | $30.4\%_{-25.3\%}$ | $45.6\%_{-21.3\%}$ | $0.3\%_{-36.0\%}$ | $29.1\%_{-16.8\%}$ | $56.9\%_{-15.5\%}$ | $60.6\%_{-25.9\%}$ | $37.1\%_{-23.5\%}$ |
| | | | Deactivate 25 neurons in total (0.006%) | | | | |
| Deactivate Random | $55.4\%_{-0.3\%}$ | $66.9\%_{-0.0\%}$ | $36.2\%_{-0.1\%}$ | $46.1\%_{+0.2\%}$ | $72.2\%_{-0.2\%}$ | $86.0\%_{-0.5\%}$ | $60.5\%_{-0.1\%}$ |
| Deactivate High-Mean | $55.0\%_{-0.7\%}$ | $67.0\%_{+0.1\%}$ | $36.2\%_{-0.1\%}$ | $46.0\%_{+0.1\%}$ | $71.9\%_{-0.5\%}$ | $85.8\%_{-0.7\%}$ | $60.3\%_{-0.3\%}$ |
| Deactivate High-$\Delta$ | $40.9\%_{-14.8\%}$ | $52.9\%_{-14.0\%}$ | $2.0\%_{-34.3\%}$ | $32.0\%_{-13.9\%}$ | $60.5\%_{-11.9\%}$ | $68.6\%_{-17.9\%}$ | $42.8\%_{-17.8\%}$ |

### B.3. Fine-Grained Ablation within NAG

To further examine which neurons within an NAG contribute most to performance, we perform a more fine-grained ablation by deactivating only 28 neurons (approximately 0.006% of all neurons), selected according to different criteria:

1) High-Mean impact: neurons selected based on the average impact score across inputs;

2) High-$\Delta$ impact: neurons selected based on the largest differences in mean impact scores between target examples and random inputs, computed over 10k HellaSwag samples and 10k random inputs during inference;

3) Random: randomly select 28 neurons from all neurons, 1 neuron per layer.

Deactivating High-$\Delta$ neurons causes a pronounced performance drop of 17.8%, while deactivating neurons selected by High-Mean impact or random sampling yields negligible degradation.

This observation provides additional mechanistic insight into why NAG is effective. Neurons with High-$\Delta$ impact scores between target examples and random inputs tend to respond selectively to different samples, making them more discriminative and task-sensitive than neurons that are uniformly activated.

## C. Validation of the Impact Score Against Loss Change

Our neuron impact score (Sec. 2) is motivated as a local approximation to avoid the cost of evaluating the change in final output. To verify that this local proxy correlates with a more behaviorally relevant quantity, we validate it against loss change on Qwen3-1.7B-Base using 500 samples.

**Setup.** For each of the up_proj neurons, we compute its impact score within each layer and rank them by impact. We then group the ranked neurons into 122 bins of 50 neurons each, and for each bin, deactivate all neurons of the same rank across layers and measure the resulting $|\Delta\text{loss}|$. Grouping is necessary because single-neuron deactivation produces near-zero loss changes, easily dominated by noise; grouping neurons of the same rank across layers produces a more stable loss-change signal.

**Results.** The Pearson correlation between the group mean impact score and $|\Delta\text{loss}|$ is $+\mathbf{0.71 \pm 0.02}$, showing strong positive correlation. Moreover, deactivating the top 0.8% neurons causes $\mathbf{159\times}$ more loss change than deactivating mid-rank neurons, providing direct evidence that the impact score correctly identifies neurons with disproportionately large effects on model behavior. This validates the impact score as an effective local proxy for the "expensive" end-to-end output change.

## D. Equivalence Between Group Similarity and Average Pairwise Similarity

In Sec. 2, we define two forms of NAG-based similarity: the pairwise similarity $\text{Sim}(c, c')$ as a Dice-style overlap, and the group similarity $\text{Sim}(c, \mathcal{D})$ as a layer-averaged frequency score. We show here that, under our setting where each sample

selects exactly $K$ neurons per layer, the group similarity is mathematically equivalent to the average of pairwise similarities:

$$\text{Sim}(c, \mathcal{D}) = \frac{1}{|\mathcal{D}|} \sum_{c' \in \mathcal{D}} \text{Sim}(c, c').$$

**Pairwise similarity decomposition.** The pairwise similarity is defined as

$$\text{Sim}(c, c') = \frac{2 \left| \text{NAG}(c) \cap \text{NAG}(c') \right|}{|\text{NAG}(c)| + |\text{NAG}(c')|}.$$

Since each sample selects exactly $K$ neurons per layer across $L$ layers, $|\text{NAG}(c)| = |\text{NAG}(c')| = L \cdot K$, so

$$\text{Sim}(c, c') = \frac{|\text{NAG}(c) \cap \text{NAG}(c')|}{L \cdot K}.$$

The intersection decomposes per layer:

$$|\text{NAG}(c) \cap \text{NAG}(c')| = \sum_{\ell=1}^{L} |N_\ell^{(K)}(c) \cap N_\ell^{(K)}(c')|,$$

giving

$$\text{Sim}(c, c') = \frac{1}{L} \sum_{\ell=1}^{L} \frac{|N_\ell^{(K)}(c) \cap N_\ell^{(K)}(c')|}{K}.$$

**Average pairwise similarity over $\mathcal{D}$.** Averaging over $\mathcal{D}$:

$$\frac{1}{|\mathcal{D}|} \sum_{c' \in \mathcal{D}} \text{Sim}(c, c') = \frac{1}{L} \sum_{\ell=1}^{L} \frac{1}{K} \sum_{k \in N_\ell^{(K)}(c)} \underbrace{\frac{1}{|\mathcal{D}|} \sum_{c' \in \mathcal{D}} \mathbf{1}[k \in N_\ell^{(K)}(c')]}_{= w_{\ell,k}(\mathcal{D})} = \frac{1}{L} \sum_{\ell=1}^{L} \frac{\sum_{k \in N_\ell^{(K)}(c)} w_{\ell,k}(\mathcal{D})}{K}.$$

**Key step: the denominator equals $K$.** The group similarity is defined as

$$\text{Sim}(c, \mathcal{D}) = \frac{1}{L} \sum_{\ell=1}^{L} \frac{\sum_{k \in N_\ell^{(K)}(c)} w_{\ell,k}(\mathcal{D})}{\sum_{k=1}^{d_\ell} w_{\ell,k}(\mathcal{D})}.$$

The denominator can be simplified:

$$\sum_{k=1}^{d_\ell} w_{\ell,k}(\mathcal{D}) = \sum_{k=1}^{d_\ell} \frac{1}{|\mathcal{D}|} \sum_{c' \in \mathcal{D}} \mathbf{1}[k \in N_\ell^{(K)}(c')] = \frac{1}{|\mathcal{D}|} \sum_{c' \in \mathcal{D}} |N_\ell^{(K)}(c')| = K,$$

since each $c'$ has exactly $K$ neurons selected per layer.

**Conclusion.** Substituting back:

$$\text{Sim}(c, \mathcal{D}) = \frac{1}{L} \sum_{\ell=1}^{L} \frac{\sum_{k \in N_\ell^{(K)}(c)} w_{\ell,k}(\mathcal{D})}{K} = \frac{1}{|\mathcal{D}|} \sum_{c' \in \mathcal{D}} \text{Sim}(c, c'),$$

which establishes the equivalence. The frequency-weighted form is simply a more efficient computation that avoids enumerating all pairs.

## E. Clustering datasets

The details of the ten datasets used for clustering experiments in Sec. 4.1.2 are shown in Tab. 10.

*Table 10.* Datasets used for task-level clustering and their corresponding task descriptions.

| Dataset | Task Description |
| --- | --- |
| ARC-Challenge (Clark et al., 2018) | Multiple-choice question answering that evaluates grade-school level science reasoning, focusing on challenging questions that require multi-step inference. |
| TriviaQA (Joshi et al., 2017) | Open-domain question answering based on trivia questions, requiring retrieval and reasoning over broad factual knowledge. |
| XStoryCloze (Lin et al., 2022) | Cross-lingual story completion task that tests narrative understanding and commonsense reasoning by selecting the most coherent story ending. |
| NQ-Open (Lee et al., 2019) | Open-ended question answering dataset derived from real user queries, requiring factual knowledge retrieval without answer candidates. |
| MathQA (Amini et al., 2019) | Mathematical problem solving involving symbolic reasoning and numerical computation, often requiring multi-step logical deduction. |
| GSM8K (Cobbe et al., 2021) | Grade-school math word problems that assess multi-step arithmetic reasoning and structured problem-solving ability. |
| XCopa (Ponti et al., 2020) | Cross-lingual causal reasoning task where the model identifies cause–effect relationships between events. |
| MuSR (Sprague et al., 2024) | Multi-step reasoning benchmark focusing on logical and compositional reasoning across multiple premises. |
| Ape210K (Zhao et al., 2020) | Large-scale dataset for instruction-following and general reasoning, covering diverse problem types and reasoning patterns. |
| XNLI (Conneau et al., 2018) | Cross-lingual natural language inference task that evaluates sentence-pair reasoning and semantic understanding across languages. |

## F. Efficiency of NAG-Based Data Selection

This section evaluates the efficiency of NAG-based data selection from two perspectives: (i) the downstream efficiency gain during pretraining, measured via compute multipliers, and (ii) the end-to-end cost of running NAG-based selection itself.

### F.1. Downstream Pretraining Efficiency

We compare the computational resources required by NAG versus other baselines to achieve the same accuracy, using compute multipliers (CM) to summarize relative efficiency (Betker, 2023; Amodei, 2025). NAG is configured by extracting width-$K = 20$ NAGs from Qwen3-1.7B-Base. A compute multiplier of $X$ between methods A and B indicates that method A requires only $1/X$ of the compute needed by method B to reach the same performance under compute-optimal training. For example, a $2\times$ compute multiplier means that one dataset achieves equivalent performance using half the training compute.

We report results across six benchmarks individually (Fig. 7a), as well as their average performance (Fig. 7b). Overall, NAG achieves an average CM improvement of 1.27–2.42$\times$ over the baselines. The gains on HellaSwag are particularly stable, ranging from 1.54–2.65$\times$, while on XStoryCloze, NAG achieves the largest improvement relative to Random, with a maximum CM of 3.7$\times$.

### F.2. End-to-end Cost of NAG-Based Selection

Beyond downstream training efficiency, the practical value of a "training-free" selection method also depends on the cost of the selection pipeline itself. The NAG selection pipeline consists of two stages: (1) *NAG extraction*, which runs a single forward pass per candidate document on the extraction model and stores the top-$K$ neuron indices; and (2) *ranking*, which computes the NAG similarity between each candidate and the target profile and selects the top-$r_f$ fraction.

**NAG extraction.** NAG extraction requires only a single forward pass per document (no generation, no backward pass) and is embarrassingly parallel across documents. For our 150B token pool with Qwen3-1.7B-Base as the extraction model, NAG extraction takes **192 GPU-hours on H100-SXM-80GB**, which is acceptable compared to the cost of model pretraining itself. Importantly, this is a *one-time* cost: once extracted, the NAG features of the candidate pool are *target-independent*

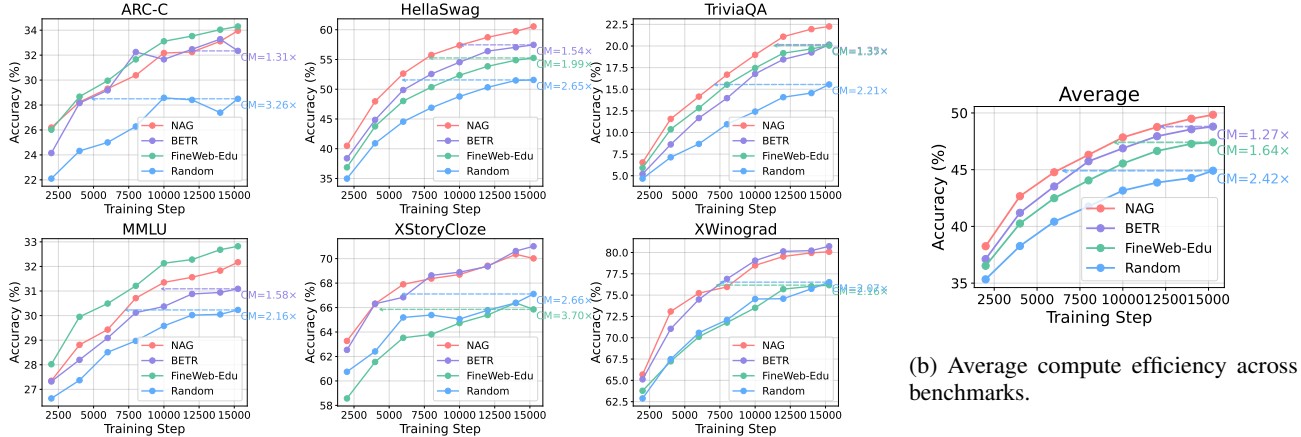

(a) Per-benchmark compute efficiency.

(b) Average compute efficiency across benchmarks.

*Figure 7.* Compute efficiency of NAG-based data selection. We report Compute multipliers (CM) of [ NAG / baseline data selection methods ] across six benchmarks, where a higher CM indicates that less compute of NAG is required to reach the same accuracy. Results are shown for (a) six benchmarks individually and (b) averaged across benchmarks.

and can be reused for any number of target tasks. In contrast, BETR requires training a new classifier and re-forwarding the entire candidate pool for each new target. Smaller extraction models can further reduce this cost: as shown in Sec. I, NAG extracted with Qwen3-0.6B-Base still outperforms all baselines, demonstrating that a smaller and cheaper extraction model suffices.

**Ranking.** Ranking is CPU-only and negligible compared to extraction. Instead of globally sorting all candidates, we estimate the top-$r_f$ filtering threshold on a small random subset of candidates and apply it to the full pool, so that each candidate only requires a single scalar comparison against the threshold. The overall ranking complexity is $O(N)$, where $N$ is the candidate pool size.

**Further cost reduction.** Beyond the choice of a smaller extraction model, additional cost reductions are possible by (1) optimizing forward-pass throughput and GPU utilization during extraction, and (2) adopting a coarse-to-fine selection scheme, where a small extraction model performs an initial filtering pass and a larger extraction model is only applied to the shortlisted candidates. We leave these directions to future work.

## G. Preliminary Analysis on the Relationship Between Task-Level Discriminability of NAG Signals and Downstream Utility

In Sec. 4.1.2, we show that NAG serves as a *task-discriminative representation*. In Sec. 4.2.3, we further observe that NAGs with different widths select data that lead to different downstream performance. Motivated by these findings, we conduct a preliminary study to examine the relationship between the two observations: *Does higher task-level discriminability of an NAG indicate that it captures richer task-level information, thereby enabling more effective data selection and yielding better downstream performance?*

We extract NAGs with different widths ($K = 5, 20, 40$) from Qwen3-1.7B-Base. Following the same setup as in Sec. 4.1.2, we first visualize the task representations using t-SNE (Fig. 8). We observe that when $K = 5$, the separability between different tasks is visibly reduced. In particular, the relative positioning of clusters—where task relevance is reflected (e.g., MathQA and GSM8K, both requiring mathematical reasoning, form closer clusters while remaining well separated from linguistic tasks such as XNLI)—is no longer preserved. This structure, which is clearly present for $K = 20$ and $K = 40$, disappears when using $K = 5$, suggesting that overly sparse NAG signals fail to capture sufficient task-level information.

To quantitatively measure clustering quality, we perform K-Means clustering based on NAG similarity and evaluate the results using standard clustering metrics, including Purity, NMI, and ARI.

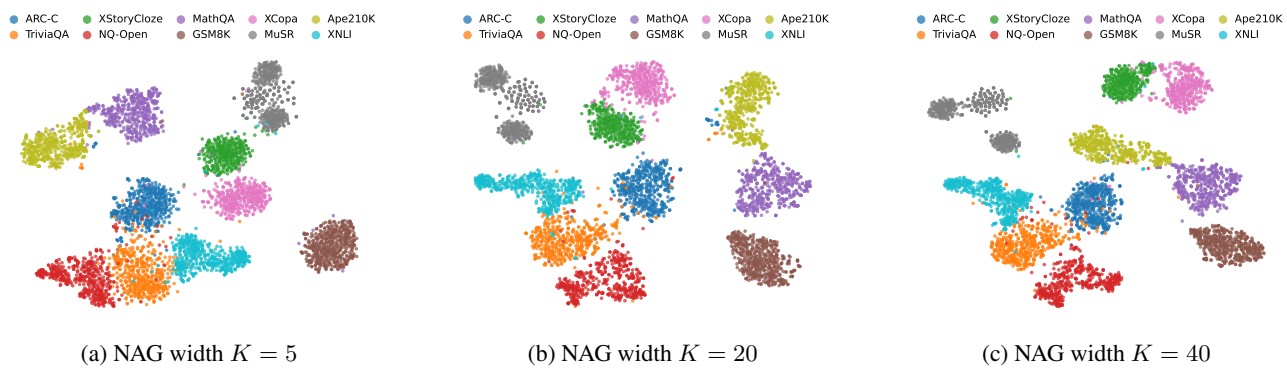

(a) NAG width $K = 5$        (b) NAG width $K = 20$        (c) NAG width $K = 40$

*Figure 8.* Task-level clustering of data instances based on NAG representations across different NAG widths. Corresponding quantitative measurement is shown in Tab. 11.

*Table 11.* Relationship between task-level discriminability of NAG signals and downstream performance. We report clustering quality metrics (Purity, NMI, ARI) obtained by K-Means clustering based on NAG similarity under different NAG widths $K$, together with the downstream accuracy on HellaSwag when using the corresponding NAG configuration for targeted data selection. Higher task-level separability of NAG representations consistently correlates with improved downstream utility of the selected data.

| NAG width $K$ | Purity ↑ | NMI ↑ | ARI ↑ | HellaSwag Acc ↑ |
|---|---|---|---|---|
| 5 | 0.878 | 0.876 | 0.797 | 58.0% |
| 20 | 0.973 | 0.945 | 0.946 | 60.6% |
| 40 | 0.963 | 0.931 | 0.920 | 60.2% |

**Clustering Evaluation Metrics.** We evaluate the alignment between unsupervised clustering structure and dataset semantics using *Purity*, *Normalized Mutual Information (NMI)* (Strehl & Ghosh, 2003), and *Adjusted Rand Index (ARI)* (Bourel et al., 2019). Purity measures cluster homogeneity and is defined as

$$\text{Purity} = \frac{1}{N} \sum_k \max_j |C_k \cap L_j|,$$

where $C_k$ denotes the $k$-th cluster, $L_j$ denotes the $j$-th dataset label, and $N$ is the total number of samples. NMI captures the global agreement between cluster assignments and dataset labels from an information-theoretic perspective:

$$\text{NMI}(C, L) = \frac{I(C; L)}{\sqrt{H(C) H(L)}},$$

where $I(\cdot; \cdot)$ is mutual information and $H(\cdot)$ denotes entropy. ARI evaluates pairwise sample consistency while correcting for chance agreement, yielding values in $[-1, 1]$, with higher scores indicating stronger alignment. Together, these metrics provide complementary assessments of local cluster purity, global partition alignment, and fine-grained structural consistency.

We then compare these metrics with the downstream performance obtained when using the corresponding NAG settings for targeted data selection on HellaSwag, as reported in Tab. 11.

Our results show a clear positive correlation: NAG configurations with higher task-level separability (i.e., higher Purity/NMI/ARI) consistently lead to higher downstream utility of the selected data. This finding provides further empirical support for the positive association between NAG-based task representation quality and downstream gains from data selection. Moreover, it offers practical guidance for selecting NAG configurations: task-level separability can serve as a lightweight proxy for downstream utility, reducing the need for repeated costly training-based validation.

## H. Sensitivity of NAG-based Selection to Target Set Size and Choice

A natural concern about target-oriented data selection methods is how sensitive the selection is to the size and choice of the target set. In this section, we analyze the sensitivity of NAG's data selection to target set size and choice on HellaSwag.

**Setup.** We sample target subsets of varying sizes ($|\mathcal{D}_{\text{target}}| \in \{200, 500, 1000, 2000, 5000\}$) from the full 10k HellaSwag target set, and use each subset to rank 1M candidate documents randomly sampled from RefinedWeb by NAG similarity. For each size, we repeat the sampling 5 times with different random seeds, producing 5 independent rankings per size. NAG is extracted from Qwen3-1.7B-Base following the default configuration.

We evaluate ranking stability along two orthogonal dimensions:

- **Intra-size (sensitivity to choice)**: consistency across 5 random draws of the same size. This measures how much the selection changes when different samples of the same size are used.

- **Cross-size (sensitivity to size)**: similarity to the full 10k baseline. This measures how much the selection changes as the target set size varies.

We report two complementary metrics: Spearman rank correlation $\rho$ over the 1M candidates, and Jaccard overlap of the top-20% selected data (i.e., top 200K candidates).

**Results.** As shown in Tab. 12, NAG-based ranking is highly robust to both target set size and choice. Spearman $\rho \geq 0.999$ across all sizes (both intra- and cross-size), indicating near-identical rankings. Even with only 200 target samples, the top-20% selected data overlaps 94% with the full 10k setting. This shows that NAG's Target Profile stabilizes with a very small number of target samples, and the data selection is effectively insensitive to which specific samples are used. In practice, this means that only around 200 in-domain samples are sufficient for effective NAG-based selection, a very low bar compared to pretraining cost.

*Table 12.* Sensitivity of NAG-based ranking to target set size and choice on HellaSwag. Intra-size metrics measure consistency across 5 random draws of the same size (sensitivity to choice); Cross-size metrics measure similarity to the full 10k baseline (sensitivity to size).

| $|\mathcal{D}_{\text{target}}|$ | Intra $\rho$ | Cross $\rho$ | Intra Jaccard | Cross Jaccard |
|---|---|---|---|---|
| 200 | 0.999 | 0.999 | $0.920 \pm 0.013$ | $0.940 \pm 0.013$ |
| 500 | 1.000 | 1.000 | $0.951 \pm 0.011$ | $0.965 \pm 0.005$ |
| 1,000 | 1.000 | 1.000 | $0.957 \pm 0.014$ | $0.970 \pm 0.010$ |
| 2,000 | 1.000 | 1.000 | $0.981 \pm 0.003$ | $0.985 \pm 0.004$ |
| 5,000 | 1.000 | 1.000 | $0.987 \pm 0.003$ | $0.989 \pm 0.003$ |

## I. Scaling Experiments

The main experiments in the paper train a 1.2B model on a 150B token pool. To verify that NAG generalizes across scale, we conduct two additional experiments that vary the training scale and the extraction model scale respectively. Both experiments use Qwen3-1.7B-Base or Qwen3-0.6B-Base as the extraction model and HellaSwag as the target under the Single-Target setting.

**Larger trained model (7B).** We scale the trained model from 1.2B to 7B and the training budget from 30B to 100B tokens selected from a 500B RefinedWeb pool. NAG is extracted using Qwen3-1.7B-Base. As shown in Tab. 13, NAG achieves a +8.4% improvement over random at 7B scale, comparable to the +9.0% improvement observed at 1.2B scale. This provides preliminary evidence that NAG's effectiveness is maintained at larger training scales.

*Table 13.* NAG performance at larger training scale. HellaSwag is used as the target. NAG is extracted from Qwen3-1.7B-Base.

| Method | Scale (Model / Token) | HellaSwag (%) |
|---|---|---|
| Random | 1.2B / 30B | 51.6 |
| NAG | 1.2B / 30B | 60.6 (+9.0) |
| Random | 7B / 100B | 63.0 |
| NAG | 7B / 100B | **71.4 (+8.4)** |

**Smaller extraction model (Qwen3-0.6B).** We further examine whether NAG is effective when the extraction model (Qwen3-0.6B) is *smaller* than the trained model (1.2B), which would be desirable for scaling NAG to larger training setups.

We extract NAG using Qwen3-0.6B-Base and train a 1.2B model on the selected 30B tokens. As shown in Tab. 14, NAG extracted from Qwen3-0.6B still outperforms all baselines on HellaSwag, demonstrating that NAG does not require the extraction model to be larger than the trained model.

*Table 14.* NAG performance with a smaller extraction model (Qwen3-0.6B, smaller than the 1.2B trained model) on HellaSwag under the Single-Target setting.

| Method | HellaSwag (%) |
|---|---|
| Random | 51.6 |
| FineWeb-Edu | 55.3 |
| BETR | 57.5 |
| NAG$_{\text{Qwen3-0.6B}}$ | **59.9** |

Together, these two experiments confirm that NAG transfers well across both extraction-model and trained-model scale gaps, supporting its practical scalability.

## J. Statistical Reliability of the Main Results

We report two complementary pieces of statistical evidence to support the reliability of our main results.

**Run-to-run variance.** To quantify the variance of our training and evaluation pipeline, we run the random baseline 5 times and report the mean and standard deviation on all six benchmarks in Tab. 15. The standard deviations are in the range 0.18%–0.55%, an order of magnitude smaller than NAG's gains across all benchmarks.

*Table 15.* Run-to-run variance of the random baseline. We report the mean and standard deviation over 5 independent pretraining runs, along with NAG's average gain across three backbone models (from Tab. 1).

| (%) | ARC-C | HellaSwag | TriviaQA | MMLU | XStoryCloze | XWinograd |
|---|---|---|---|---|---|---|
| Random (mean) | 28.75 | 51.44 | 15.12 | 30.06 | 66.62 | 76.50 |
| Random (std) | 0.51 | 0.28 | 0.40 | 0.18 | 0.43 | 0.55 |
| NAG gain | +6.2 | +8.1 | +6.5 | +1.4 | +3.3 | +3.9 |

**Evaluation standard errors.** For each reported accuracy, we also report the binomial standard error (computed as $\sqrt{p(1-p)/n}$, where $n$ is the test set size; see Tab. 6 for test sizes). We report full standard errors for all main tables in the paper below. NAG's gains consistently exceed the standard error range across all benchmarks, confirming that the reported improvements are statistically reliable.

*Table 16.* Standard errors for Tab. 1 (main results).

| Method | ARC-C | HellaSwag | TriviaQA | MMLU | XStoryCloze | XWinograd |
|---|---|---|---|---|---|---|
| Random | $28.5 \pm 1.3$ | $51.6 \pm 0.5$ | $15.6 \pm 0.3$ | $30.2 \pm 0.4$ | $67.1 \pm 1.2$ | $76.5 \pm 0.9$ |
| FineWeb-Edu | $34.3 \pm 1.4$ | $55.3 \pm 0.5$ | $20.1 \pm 0.3$ | $32.8 \pm 0.4$ | $65.9 \pm 1.2$ | $76.2 \pm 0.9$ |
| | | | Single-Target | | | |
| BETR | $32.3 \pm 1.4$ | $57.5 \pm 0.5$ | $20.2 \pm 0.3$ | $31.1 \pm 0.4$ | $71.0 \pm 1.2$ | $80.7 \pm 0.8$ |
| NAG$_{\text{Qwen3-1.7B}}$ | $34.0 \pm 1.4$ | $60.6 \pm 0.5$ | $22.3 \pm 0.3$ | $32.2 \pm 0.4$ | $70.0 \pm 1.2$ | $80.1 \pm 0.8$ |
| NAG$_{\text{Llama-3.2-3B}}$ | $35.0 \pm 1.4$ | $58.6 \pm 0.5$ | $21.3 \pm 0.3$ | $31.5 \pm 0.4$ | $70.8 \pm 1.2$ | $80.6 \pm 0.8$ |
| NAG$_{\text{SmolLM3-3B}}$ | $35.0 \pm 1.4$ | $59.8 \pm 0.5$ | $22.6 \pm 0.3$ | $31.2 \pm 0.4$ | $70.5 \pm 1.2$ | $80.6 \pm 0.8$ |
| | | | Multi-Target | | | |
| BETR | $30.3 \pm 1.3$ | $49.3 \pm 0.5$ | $11.6 \pm 0.2$ | $29.9 \pm 0.4$ | $69.5 \pm 1.2$ | $76.1 \pm 0.9$ |
| NAG$_{\text{Qwen3-1.7B}}$ | $33.4 \pm 1.4$ | $57.8 \pm 0.5$ | $19.2 \pm 0.3$ | $31.5 \pm 0.4$ | $69.3 \pm 1.2$ | $79.9 \pm 0.8$ |
| NAG$_{\text{Llama-3.2-3B}}$ | $32.0 \pm 1.4$ | $54.9 \pm 0.5$ | $18.0 \pm 0.3$ | $31.4 \pm 0.4$ | $69.8 \pm 1.2$ | $79.9 \pm 0.8$ |
| NAG$_{\text{SmolLM3-3B}}$ | $31.8 \pm 1.4$ | $55.2 \pm 0.5$ | $19.9 \pm 0.3$ | $30.6 \pm 0.4$ | $69.2 \pm 1.2$ | $80.2 \pm 0.8$ |

*Table 17.* Standard errors for Tab. 2 (NAG combined with FineWeb-Edu).

| Method | ARC-C | HellaSwag | TriviaQA | MMLU | XStoryCloze | XWinograd |
|---|---|---|---|---|---|---|
| FineWeb-Edu | $34.3 \pm 1.4$ | $55.3 \pm 0.5$ | $20.1 \pm 0.3$ | $32.8 \pm 0.4$ | $65.9 \pm 1.2$ | $76.2 \pm 0.9$ |
| + NAG$_{\text{Qwen3-1.7B}}$ | $35.3 \pm 1.4$ | $57.7 \pm 0.5$ | $21.7 \pm 0.3$ | $32.5 \pm 0.4$ | $67.2 \pm 1.2$ | $79.2 \pm 0.8$ |
| + NAG$_{\text{Llama-3.2-3B}}$ | $35.2 \pm 1.4$ | $57.4 \pm 0.5$ | $21.7 \pm 0.3$ | $32.7 \pm 0.4$ | $68.2 \pm 1.2$ | $78.6 \pm 0.9$ |
| + NAG$_{\text{SmolLM3-3B}}$ | $35.7 \pm 1.4$ | $58.1 \pm 0.5$ | $22.7 \pm 0.3$ | $33.1 \pm 0.4$ | $69.0 \pm 1.2$ | $78.9 \pm 0.8$ |

*Table 18.* Standard errors for Tab. 4 (All-Layer vs Last-Layer NAG).

| Method | ARC-C | HellaSwag | TriviaQA | MMLU | XStoryCloze | XWinograd |
|---|---|---|---|---|---|---|
| NAG$_{\text{All Layer}}$ | $34.0 \pm 1.4$ | $60.6 \pm 0.5$ | $22.3 \pm 0.3$ | $32.2 \pm 0.4$ | $70.0 \pm 1.2$ | $80.1 \pm 0.8$ |
| NAG$_{\text{Last Layer}}$ | $30.5 \pm 1.3$ | $55.2 \pm 0.5$ | $15.1 \pm 0.3$ | $29.9 \pm 0.4$ | $67.8 \pm 1.2$ | $75.5 \pm 0.9$ |

