# OpenReview forum: "Target-Oriented Pretraining Data Selection via Neuron-Activated Graph"
_ICML.cc/2026/Conference — ICML 2026 regular_

### Official Review · Reviewer_VZo4 · 2026-03-08

**Soundness:** 4
**Presentation:** 4
**Significance:** 3
**Originality:** 3
**Overall Recommendation:** 4
**Confidence:** 3

**Summary:**

The paper proposes a novel data selection method that evaluates the data effectiveness to the target task via the neuron activation patterns (Neuron Activate Graph in the paper) of LLMs. The paper first gathers the NAGs of the target task and the candidate data, and then compares the NAGs to select the data that has similar activation patterns to the target task. The paper conducts comprehensive experiments on 6 different tasks and validated its effectiveness. Additionally, the paper extensively evaluates the ablated components of the method and provides insightful analysis on the results. The paper is well-written and easy to follow.

**Compliance With Llm Reviewing Policy:**

Affirmed.

**Final Justification:**

The rebuttal content solves most of my concerns. And the experiments showing that small model conclusions are applicable to larger models enhance the quality. Yet the cost for data selection is sitll a minor concern to me.

I would keep the acceptance recommendation.

**Key Questions For Authors:**

1. The baseline setting is not clear enough. What is the backbone LLM used in the baseline experiments?
2. It is expected that the paper provides more theoretical analysis to explain why the method works. E.g., you may start by defining the data selection problem, and then show how the NAG fits into the problem and why it can solve the problem.

**Limitations:**

The suggested limitations are listed in the weaknesses and questions.

**Strengths And Weaknesses:**

**Strength**
1. The experiments are comprehensive and the results are solid.
2. The proposed method is simple and widely applicable. Could be impactful to the community.
3. The paper has a clear presentation, Fig.2 greatly explains the whole idea.

**Weakness**
1. The Sec. 4.1 explains the function of the method in an ad-hoc manner. For example, in Sec. 4.1.1, since the neurons are selected by their activation patterns, suppressing the neurons of course leads to a decrease of the performance.
2. From the presentation, it seems that the whole LLM is used to build the NAG, the process is conducted at the sample level. This raises the concern of the cost of the method, especially for larger LLMs. It would be better if the paper can provide more details on the cost and how to reduce it.

---

> ### Author Rebuttal · Authors · 2026-03-30
>
> We thank the reviewer for the constructive feedback.
>
> ### W1: Deactivation experiment
>
> Neurons selected by activation patterns do **not** necessarily lead to performance decrease when suppressed. Appendix Table 8 provides a controlled comparison: deactivating only 28 neurons (approximately 0.006% of all neurons), neurons with the highest mean activation (High-Mean) cause only a **0.3%** drop, while neurons with the largest target-vs-random impact difference (High-Δ) cause a **17.8%** drop. This directly shows that high activation does not imply task importance. Precisely because this connection is not obvious, the deactivation experiment is necessary to validate that NAG-selected neurons are indeed functionally critical for the target task.
>
> ### W2: Cost of NAG construction / Ranking
>
> NAG extraction requires only a single forward pass per document (no generation needed), and is embarrassingly parallel. It takes **192 GPU-hours (H100-SXM-80GB)** for 150B tokens with Qwen3-1.7B, acceptable compared to model pretraining cost. Once extracted, NAGs are reusable for any target—unlike BETR which requires retraining a classifier and re-forwarding the entire pool per target. Smaller extraction models can further reduce this cost (e.g., Qwen3-0.6B still outperforms all baselines, see Reviewer Jkcp W3). Further extraction cost reduction is possible by (1) optimizing forward pass throughput and GPU utilization, and (2) coarse-to-fine selection using a small extraction model for initial filtering followed by a larger model on the shortlist.
>
> Ranking is CPU-only and negligible: we estimate the filtering threshold on a small subset and apply it to the full pool in O(N).
>
> ### Q1: Backbone LLM in baseline experiments
>
> As stated in Sec. 3.3 (L200-208), all methods (Random, FineWeb-Edu, BETR, and NAG) train the **same** 1.2B model architecture from random initialization, with identical optimization settings and a fixed budget of 30B tokens. The only difference lies in the data selection strategy. We note that the backbone models listed in Table 1 (e.g., Qwen3-1.7B) refer to the off-the-shelf models used for **NAG extraction**, not the model being trained.
>
> ### Q2: More theoretical analysis
>
> We offer the following high-level framing:
>
> The data selection problem can be formulated as: given a candidate pool $\mathcal{C}$ and target task $\mathcal{T}$, select $\mathcal{S} \subset \mathcal{C}$ that maximizes downstream performance on $\mathcal{T}$. NAG builds on the mechanistic interpretability finding that LLMs develop specialized neuron circuits for different capabilities [5][6]. Based on this, NAG selects data that activates similar neuron circuits as the target examples, thereby prioritizing data that exercises the same model capabilities.
>
> In Section 4, we provide empirical analyses supporting this framework from two perspectives: **why NAG works** (the identified neurons are task-critical (Table 3) and task-discriminative (Fig. 3, Table 10)) and **how NAG operates** (aggregating signals across layers (Table 4) at sparse neuron ratios (Fig. 6)). We will add a more formal problem statement in the revision.
>
> We hope this addresses the reviewer's concerns and are happy to discuss further.
>
> ### References
>
> [1] DataComp-LM: In search of the next generation of training sets for language models
>
> [2] Predictive Data Selection: The Data That Predicts Is the Data That Teaches
>
> [3] Improving Pretraining Data Using Perplexity Correlations
>
> [4] Optimizing pre-training via target-aware source data selection
>
> [5] Task-Specific Skill Localization in Fine-tuned Language Models
>
> [6] How do large language models handle multilingualism?

---

> > ### Author Rebuttal · Reviewer_VZo4 · 2026-04-02
> >
> > 1. The experiments do show that the proposed neuron selection method can best suppress the model performance. However, the conclusion is not changed; your method is designed to find the neurons that can best alter the output, and the experiments in Sec. 4.1 is a validation of your method, but it can not provide an explanation of why the method works. In other words, this section should provide the causality between the impactful neurons and the performance gain from the selected data, not showing that the selected neurons are indeed impactful. You may consider changing the interval of the selected neurons to show when the neurons are less and less impactful, the selected data are less and less effective.
> >
> > 2. In the pretraining stage, the model is normally trained only for a few epochs. Thus, spending an epoch only for selecting data still doesn't make it worth it. However, the point of using a smaller model to select data and apply the conclusion to larger models is a good idea. It is expected to see more extension of this idea in the experiments.
> >
> > 3. Additionally, I notice that the concrete neuron selection method is missing, how do you calculate the top-k impactful neurons of a layer? Finding the best combo of k neurons from all neurons seems to be a combinatorial optimization problem, which is hard to solve.

---

> > > ### Author Response · Authors · 2026-04-02
> > >
> > > We thank the reviewer for the thoughtful follow-up questions.
> > >
> > > 1. **Causality between impactful neurons and data selection**: The deactivation experiment (Sec. 4.1.1) was designed to validate that the neurons NAG selects are genuinely task-critical, which is the foundation of NAG's data selection — if the selected neurons were not important for the task, using them to measure data-task similarity would lack justification. Thank you for this insightful suggestion. We agree that directly varying neuron impactfulness and observing the resulting data selection quality would strengthen the causal argument, which we leave as future work.
> > >
> > > 2. **Cost**: Thank you for raising this point. We may have not explained this clearly enough—NAG extraction is a **single forward pass** per document, not a training epoch. There is no backward pass or parameter update, only collecting activation patterns during inference. This makes it significantly cheaper than training. If we have misunderstood the reviewer's concern, we are happy to continue the discussion.
> > > Regarding using a smaller model to select data for larger models—thank you for the positive feedback on this direction. Our current results show this works across two scale gaps: Qwen3-0.6B extraction for 1.2B training (see Reviewer Jkcp W3) and Qwen3-1.7B extraction for 7B training (see Reviewer Jkcp W1). We agree this is a promising direction worth further exploration.
> > >
> > > 3. **Neuron selection**: As described in Sec. 2.2 (L116-122), the top-K selection is not combinatorial optimization. For each neuron $k$, we independently compute its impact score $\|h_{in}^\top W_{:,k}\|_2$ (Sec. 2.1), then simply sort and take the top-K per layer. The complexity is $O(d \log d)$ per layer (just sorting). If we have misunderstood the reviewer's question, please let us know.

---

### Official Review · Reviewer_Jkcp · 2026-03-09

**Soundness:** 3
**Presentation:** 3
**Significance:** 2
**Originality:** 3
**Overall Recommendation:** 4
**Confidence:** 3

**Summary:**

This paper introduces Neuron-Activated Graph data selection (NAG-based selection) and conducts related experiments with pre-training of Language Models. In this work, a NAG is a data structure that, for a given sample, contains the indices of the top-K most activated neurons per layer.

The work builds on recent literature advocating for target-aware data selection, where only a subset of pre-training data is retained for training based on its estimated relevance to a downstream task of interest. Specifically, NAG-based selection retains samples based on the similarity between NAGs computed on pre-training samples and those computed on some target data (e.g., the training set of a benchmark). The similarity between NAGs is computed differently based on whether (a) two inputs are being compared or (b) an input is being compared to a group.

Experiments are conducted on 1.2B-sized Language Models trained on a fixed 30B token budget selected from a 150B random subset of RefinedWeb, evaluating on 6 common benchmarks (MMLU, HellaSwag, TriviaQA, ARC-C, XWinograd and XStoryCloze). Results show that NAG performs better than BETR (a recent embedding-based approach for target-aware data selection) and FineWeb-Edu's "educational" classifier, which instead implies a universal notion of quality.

Among other findings, follow-up analyses show that (i) NAG can be jointly used with general quality filters, (ii) the neurons selected by NAG lead to severe performance degradation if zeroed out, (iii) up-projections in feedforward networks within transformer blocks lead to the best data selection by a large margin.

**Compliance With Llm Reviewing Policy:**

Affirmed.

**Final Justification:**

The rebuttal addresses two of my primary concerns (scale and small-to-large transfer for scalability) and shows that the improvement brought by NAGs is definitely not attributable to noise in the experimental evaluation.

Some aspects remain a concern (e.g., W2 and I still believe there would have been better ways to convey the message that Table 3 seeks to convey), but overall they are minor compared to the merits of the paper.

At the current stage, I would therefore recommend accepting this work.

**Key Questions For Authors:**

1. What is the author's perspective on scale? Do they agree/disagree about the fact that this paper might benefit from experiments at larger compute budgets (e.g., more tokens and/or bigger model size)?
2. Did the authors explore alternative methods that aggregate layer-wise features? Do the authors have evidence that using a "graph-like" representation is better than using features, when both alternatives are evaluated with the same per-layer aggregation scheme in mind?
3. Do the authors have feedback or thoughts about how to scale NAG-based selection to larger corpora? What is the expected time to collect NAGs for a data pool compared to BETR as the amount of pre-training data scales?
4. How would the proposed method behave when training a larger model while scoring documents with a smaller model to maintain a minimal degree of scalability?
5. (Minor) How did the authors select the hyperparameters of their NAG-based selection variants? Could they report their hyperparameter selection in detail?
6. (Minor) If the authors did conduct multiple runs, could they report the observed standard deviation (even for the random baseline)? This would be beneficial to understand the variability of the evaluation spectrum and better isolate performance gains from noise.

**Limitations:**

The authors reported a minimal "Impact Statement" following ICML's original template. While I agree with their assessment, I believe this paper would benefit from a dedicated "Limitations & Future Work" section discussing future directions to explore and/or hints to improve on the proposed data selection strategy.

**Strengths And Weaknesses:**

**Strengths.**
- The paper focuses on data selection for Language Model pretraining, which is an important problem with both efficiency and utility implications.
- The writing is clear, which makes the understanding of the paper easy.
- The proposed method is interesting. Using neuron activations for data selection is, in my opinion, a nice way to transfer knowledge from other fields, such as explainability, model pruning, and parameter-efficient fine-tuning, where similar "activation-based" criteria have been used for different purposes.
- The results seem to consistently support the proposed method, with NAG outperforming the recently proposed BETR, which shares the underlying goal of target-aware data filtering.

**Weaknesses.**
- I think this paper is crucially missing experiments at larger scales. The current manuscript reports experiments with 1.2B-sized Language Models only, but there's evidence that data curation methods are compute-dependent [a] and therefore might not transfer to larger models and token budgets. In my opinion, since this is a pre-training paper, the scale of the experiments is a bit too modest.
- While I do believe that the proposed method is interesting, the manuscript does not report the necessity of using "graphs" instead of feature embeddings. Specifically, this work suggests that NAGs outperform BETR-like approaches because they aggregate information from different layers instead of only using last-layer information. However, this does not directly justify using graphs, i.e., one could equivalently use embedding-based selection aggregating from different layers.
- The proposed NAG-based selection is not designed to scale well, which is instead a core desideratum for pretraining data selection. Forwarding a model and computing similarities over a larger data pool (e.g., over DCLM [b]) would make the method computationally prohibitive, in contrast to BETR, which distills a lightweight classifier with scalability in mind. In my opinion, this is also critically connected to the earlier point of missing experiments at larger scales. Specifically, there's evidence that data filtering models should have larger capacity than trained models [c]. This hints that scaling the proposed NAG-based selection might need a larger model to rank pretraining documents, thereby becoming increasingly more prohibitive with scale.

**Minor Point.**
- I am a bit confused by the experiment reported in Table 3. The manuscript mentions that *"[...] we evaluate whether the neurons selected by NAG are indeed crucial to impact the model’s final performance by selectively deactivating them"* (lines 325-327). The experiment seems to be evaluating Qwen3-1.7B-Base directly, since the results are different from those reported in Tables 1&2. I would have expected results of NAG-based selection *without* the top-K activated neurons instead. In its current form, Table 3 does not add much to the existing literature, which shows that, for some tasks, only a subset of parameters is relevant (e.g., [d]).

**References**\
[a] Goyal, Sachin, et al. "Scaling Laws for Data Filtering--Data Curation cannot be Compute Agnostic." Proceedings of the IEEE/CVF Conference on Computer Vision and Pattern Recognition. 2024.\
[b] Li, Jeffrey, et al. "Datacomp-lm: In search of the next generation of training sets for language models." Advances in Neural Information Processing Systems 37 (2024): 14200-14282.\
[c] Udandarao, Vishaal, et al. "Active data curation effectively distills large-scale multimodal models." Proceedings of the Computer Vision and Pattern Recognition Conference. 2025.\
[d] Nguyen, Bac, et al. "Saft: Towards out-of-distribution generalization in fine-tuning." European Conference on Computer Vision. Cham: Springer Nature Switzerland, 2024.

---

> ### Author Rebuttal · Authors · 2026-03-30
>
> We thank the reviewer for the constructive feedback.
>
> ### W1 & Q1: Missing experiments at larger scales
>
> We agree that experiments at larger training scales would further strengthen the paper. We chose 1.2B to enable comprehensive ablations (Fig. 4-6, Tables 1-4) given the high cost of pretraining.
>
> We have now completed a **7B model experiment** (training on 100B tokens selected from a 500B RefinedWeb pool, with HellaSwag as the target).
>
> | Method | Scale (Model/Token) | HellaSwag (%) |
> | --- | --- | --- |
> | Random | 1.2B / 30B | 51.6 |
> | NAG_{Qwen3-1.7B} | 1.2B / 30B | 60.6 (+9.0) |
> | Random | 7B / 100B | 63.0 |
> | NAG_{Qwen3-1.7B} | 7B / 100B | **71.4 (+8.4)** |
>
> The improvement from NAG at 7B scale (+8.4%) is comparable to that at 1.2B scale (+9.0%), providing preliminary evidence that NAG's effectiveness is maintained at larger training scales.
>
> ### W2 & Q2: Graph vs aggregated embeddings
>
> This is a valuable question. NAG deliberately uses discrete neuron index sets rather than continuous embeddings, and there are several reasons for this design:
>
> 1. **Robustness**: NAG only records **which** neurons are activated, not their exact values, making it inherently robust to activation noise. Continuous embedding similarity is more susceptible to noise in high dimensions.
> 2. **Interpretability**: Aggregating embeddings across a target set (e.g., by averaging) produces an opaque high-dimensional vector. NAG instead aggregates by counting **neuron occurrence frequency** across target samples (Sec. 2.3), producing a directly interpretable Target Profile.
> 3. **Computational cost**: NAG compresses each layer's information into only K neuron indices (K ≈ 0.3% of the embedding dimension), making storage and similarity computation orders of magnitude cheaper than full multi-layer embeddings—especially when processing 150B+ tokens.
>
> ### W3 & Q3: NAG scalability to larger corpora
>
> Regarding [c] suggesting that filtering models should have larger capacity than trained models: this concern applies more directly to BETR and other classifier-based methods (e.g., [1][2][3]), which distill signals into **lightweight classifiers**—precisely the opposite of [c]'s recommendation—yet still work well in practice. NAG leverages an off-the-shelf LLM directly, and we show that NAG works even when the extraction model is smaller than the trained model: **Qwen3-0.6B** extraction for 1.2B training (below), and **Qwen3-1.7B** extraction for 7B training (see W1). Both outperform all baselines:
>
> | Method | HellaSwag (%) |
> | --- | --- |
> | Random | 51.6 |
> | FineWeb-Edu | 55.3 |
> | BETR | 57.5 |
> | **NAG_{Qwen3-0.6B}** | **59.9** |
>
> Regarding computational cost: NAG extraction cost is acceptable compared to model pretraining cost. See Reviewer VZo4 W2 response for details.
>
> ### W4 (Minor): Table 3 experiment design
>
> The "model" in L325-327 refers to the extraction model (Qwen3-1.7B-Base), from which NAG's neurons are identified. Table 3 evaluates whether these neurons are critical to this model's performance. Regarding the difference from [d]: prior work shows that task-relevant neuron subsets exist, but does not validate whether **NAG's criterion** finds them. Table 3 fills this gap.
>
> The reviewer's suggested ablation would provide end-to-end validation. Our approach decomposes this into two steps — (1) Table 3: NAG-selected neurons are task-critical; (2) Tables 1-2: data selected by these neurons improves training — explaining **why** NAG works, not just whether.
>
> ### Q4: Cross-scale transfer
>
> See our responses to W3 (Qwen3-0.6B extraction for 1.2B training) and W1 (Qwen3-1.7B extraction for 7B training). Both confirm NAG transfers well across extraction-training scale gaps.
>
> ### Q5: Hyperparameter selection
>
> Our hyperparameters are reported in Sec. 3.3, with selection rationale in Sec. 4.2 (Fig. 4-6, Table 4). All hyperparameters are held fixed across all benchmarks and settings unless otherwise specified (L210).
>
> ### Q6: Standard deviation
>
> Our main results are based on single runs due to the high cost of pretraining experiments. To quantify the variance of our training and evaluation pipeline, we run the **random baseline 5 times** and report the mean and standard deviation:
>
> | (%) | ARC-C | HellaSwag | TriviaQA | MMLU | XStoryCloze | XWinograd |
> | --- | --- | --- | --- | --- | --- | --- |
> | Random (mean) | 28.75 | 51.44 | 15.12 | 30.06 | 66.62 | 76.50 |
> | Random (std) | 0.51 | 0.28 | 0.40 | 0.18 | 0.43 | 0.55 |
> | NAG gain | +6.2 | +8.1 | +6.5 | +1.4 | +3.3 | +3.9 |
>
> The standard deviations are an order of magnitude smaller than NAG's gains across all benchmarks, confirming that our reported improvements are well above the noise level.
>
> We hope this addresses the reviewer's concerns and are happy to discuss further. (References: see Reviewer VZo4 response.)

---

> > ### Author Rebuttal · Reviewer_Jkcp · 2026-04-03
> >
> > Dear Authors,
> >
> > Thank you for your responses. My major concerns were resolved, specifically:
> > - W1 & Q1: The large-scale experiments;
> > - W3 & Q3: The small-to-large filtering experiments;
> >
> > Resolving the point about the standard deviation also plays a crucial role in this rebuttal.
> >
> > W2&Q2 remains a concern. However, I think it is acceptable if any concrete evidence or comparison between NAGs or per-layer embedding aggregation is provided in future work.
> >
> > I'd recommend completing any ongoing experiments and finalizing answers to W1&W2 to include in any revision of this work. I am confident they would improve the overall quality and significance of the manuscript.
> >
> > Best,\
> > Reviewer `Jkcp`

---

> > > ### Author Response · Authors · 2026-04-03
> > >
> > > Dear Reviewer Jkcp,
> > >
> > > Thank you very much for taking the time to carefully re-evaluate our responses and for the encouraging feedback. We truly appreciate the thoughtful and constructive suggestions you have provided throughout this review process — they have been very helpful in improving our work.
> > >
> > > We will follow your recommendation to complete the ongoing experiments and incorporate the finalized results for W1 and W2 in the revised manuscript. Thank you again for your support and guidance.
> > >
> > > Best,
> > >
> > > The Authors

---

### Official Review · Reviewer_TZ3v · 2026-03-13

**Soundness:** 3
**Presentation:** 4
**Significance:** 3
**Originality:** 3
**Overall Recommendation:** 4
**Confidence:** 4

**Summary:**

This study intends to examine an important aspect of target-oriented LM pretraining: how to choose pretraining data that better matches a desired downstream capability using neuron-level signals rather than embeddings or generic quality heuristics. The paper proposes Neuron-Activated Graph Ranking (NAG), which measures neuron impact in an off-the-shelf LLM, retains the top-K high-impact neurons per layer as a sparse signature, aggregates target examples into a profile, and ranks candidate documents by NAG similarity. Experiments pretraining a 1.2B-parameter model on 30B selected tokens from a 150B RefinedWeb pool across six benchmarks report average gains of 4.9% over random selection in the single-target setting, plus improvements over BETR and FineWeb-Edu, with supplementary analyses on deactivation, clustering, layer choice, and sparsity.

**Compliance With Llm Reviewing Policy:**

Affirmed.

**Key Questions For Authors:**

1. **Neuron impact computation at document level**

   In **Sec. 2.1**, how exactly is neuron impact computed for a full text input and then lifted to the **document level** used for ranking in **Sec. 3.1**? The equation is written for a single vector \( h_{in} \), but the actual data unit is a **document from a 150B-token pool**, with **max sequence length 2048** in Appendix A.1.

   Please specify:
   - whether impacts are **aggregated across all token positions**,
   - over **multiple chunks**,
   - or taken from a **designated position**.

   Also clarify whether this **local proxy** was validated against a more behaviorally relevant measure such as **output/logit/loss change** on a subset.

   A precise and validated answer would substantially increase my confidence in the **technical soundness and reproducibility** of the method.

2. **Single-run vs multi-seed results**

   Are the main results in **Tables 1–4** based on **single pretraining runs**, or **averages over multiple seeds**?

   I could not find:
   - standard deviations
   - confidence intervals
   - significance tests

   Since several gains over strong baselines are **modest in absolute terms**, especially outside **HellaSwag**, a **multi-seed analysis** would materially affect how strongly I interpret the empirical advantage.

   If the gains remain **stable under repeated runs**, my confidence in the empirical claims would increase significantly.

3. **End-to-end computational cost of NAG selection**

   **Fig. 7** reports compute multipliers from training curves, but what is the **end-to-end cost of NAG-based selection itself**, relative to **BETR** and **FineWeb-Edu**?

   In particular, I would like to know the **one-time cost** of:
   - extracting NAGs
   - ranking the **150B-token pool**

   Please include details on:
   - hardware used
   - throughput
   - GPU-hours

   This matters because the practical value of a **“training-free” method** depends not only on downstream pretraining efficiency but also on **selection-time cost**.

   If the preprocessing overhead is **modest or clearly amortized**, that would strengthen the paper’s efficiency claim.

4. **Target-set construction and decontamination protocol**

   In **Sec. 3.3**, target examples are said to come from **benchmark training splits**, with **decontamination against benchmark test sets**, but I could not find:

   - the exact **size of \( D_{\text{target}} \)** per benchmark
   - the **sampling protocol**
   - the **contamination-matching rules**

   Please provide these details.

   A rigorous answer here would:
   - reduce concern about **leakage**
   - improve **reproducibility**

   If the protocol is weak or underspecified, it would reduce confidence in the **target-oriented evaluation**.

**Limitations:**

The dedicated **Impact Statement** is extremely brief and effectively says that **no specific societal consequences need highlighting**, but I think the paper would benefit from a more explicit discussion of both **methodological limits** and **possible misuse**.

Constructively, I would encourage the authors to add a short **limitations subsection** covering at least the following points.

## Empirical scope limits
The study uses:
- one English source corpus (**RefinedWeb**)
- one pretraining architecture/scale (**1.2B**)
- one **30B-token training budget**
- a fairly simple **multi-target mixture** without re-weighting or deduplication.

The paper should state more explicitly that generalization to:
- other **corpora**
- other **languages**
- different **model scales**
- more realistic **multi-objective settings**

remains **unverified**.

## Statistical and reproducibility limits
The paper does not report:

- multi-seed variance
- significance tests
- exact target-set sizes
- detailed decontamination rules
- the full end-to-end cost of ranking

These omissions should be **acknowledged as current limitations** rather than left implicit.


## Potential societal risks
Target-oriented selection could:

- inherit or amplify **biases** present in the target examples
- reflect narrow **demographic, topical, or linguistic coverage**

More broadly, methods that make **model specialization easier** can also be used to optimize models for:

- narrowly **persuasive**
- **exclusionary**
- or otherwise **harmful applications**

A brief discussion of these risks would strengthen the impact section.

## Suggested mitigations
A stronger paper would describe **practical safeguards**, such as:

- auditing the **selected data distribution**
- checking for **demographic or topical skew**
- screening for **harmful content** before and after filtering
- using **diversity constraints** during selection
- validating that improvements on the **target task** do not come with disproportionate regressions on **broader or sensitive benchmarks**

A **candid discussion** of these issues would improve the paper.

**Strengths And Weaknesses:**

# Strengths

## Problem framing and conceptual clarity
- The method is grounded in an interpretable hypothesis—that inputs sharing a capability will activate similar sparse neuron subsets—rather than in an opaque auxiliary classifier, which gives the paper a distinctive conceptual angle within its own related-work framing.
- The paper explicitly separates “why NAG works” from “how NAG operates,” and this organization makes the analysis section easier to follow than a single undifferentiated ablation block.

## Controlled experimental setup and useful implementation detail
- All compared methods use the same **1.2B architecture**, **optimization recipe**, and **30B-token training budget**, so the main comparisons are reasonably controlled around the data-selection variable of interest.
- The benchmark appendix reports **task type, number of shots, metric, and short description** for all six evaluations, which helps readers reconstruct the evaluation protocol faithfully.

---

# Weaknesses

## Mathematical formulation and notation
- The impact score is motivated as a local approximation because evaluating final-output change is “expensive,” but no validation is provided showing that this local proxy correlates with a closer-to-behavior quantity such as **loss change, logit change, or gradient-based importance** (Sec. 2.1).
- The object called a **“Neuron-Activated Graph”** is formally defined as layer-wise top-\(K\) sets or as a set of layer-neuron pairs; without explicit edges or adjacency, the mathematical object currently reads more like a **sparse profile than a graph** (Sec. 2.2).
- Pairwise similarity \( \text{Sim}(c,c') \) is a **Dice-style overlap**, whereas group similarity \( \text{Sim}(c,D) \) becomes a **layer-averaged frequency score**; for why this second normalization preserves the same geometry or ranking behavior as the first definition (Sec. 2.3).
- Several notation choices reduce readability, including the use of **TopKK**, mixed \(l/\ell\) symbols in the group profile, and
  \( \|h_{in}^{\top}W_{:,k}\|_2 \) for what appears to be a scalar contribution, which makes the derivation harder to parse than necessary (Sec. 2.1–2.3).

## Empirical rigor and narrative consistency
- Some narrative claims need auditing against the tables:
  - The abstract mentions a **5.3% HellaSwag gain over “state-of-the-art baselines”**, while Table 1 shows **+3.1 over BETR** and **+5.3 over FineWeb-Edu**.
  - Sec. 3.6 also gives an **ARC-C example (35.4 > 34.7 > 34.3)** that does not match Tables 1–2 (Abstract; Sec. 3.6; Tables 1–2).
- The claim that **NAG rankings have a “high correlation” with target utility** is supported mainly by **one filtering-rate sweep on HellaSwag** and is not accompanied by an explicit **correlation coefficient across tasks or runs** (Sec. 4.1.3; Fig. 4).
- Sec. 4.1.2 relies on a **t-SNE visualization** to argue task discriminability, but the main-text setting does not report **direct quantitative clustering metrics**; those appear only later for a different width study (Sec. 4.1.2; Fig. 3; p. 15–16; Table 10).
- The appendix states a **“causal relationship”** between task-level discriminability and downstream gains, but the evidence is a **small comparison across \(K \in \{5,20,40\}\)** on HellaSwag rather than an intervention isolating discriminability itself (p. 15–16; Fig. 8; Table 10).

## Scope of validation and breadth of claims
- All candidate pretraining data come from a **single English web corpus**, **RefinedWeb** downsampled to **150B tokens**, so evidence for transfer to other source distributions remains limited (Sec. 3.1).
- All trained models use **one 1.2B-parameter architecture** and **one 30B-token budget**, which controls confounds but leaves open whether the effect size changes materially at **smaller or larger scales** (Sec. 3.3; Table 5).
- The benchmark suite is broad within **reasoning and commonsense**, but it does not test the motivating applied domains named in the introduction, such as **education, medicine, or research domains**.
- The **multi-target setup** is explicitly a **simple equal-budget mixture without re-weighting or de-duplication**, so the practical claim for real-world multi-objective scenarios should be interpreted as **preliminary rather than definitive** (Sec. 3.3; Sec. 3.5).
- The **impact statement** is extremely brief and says that **no societal consequences need highlighting**, leaving unaddressed whether **target-oriented filtering could amplify biases or over-specialize models toward small target sets**.

---

> ### Author Rebuttal · Authors · 2026-03-30
>
> We thank the reviewer for the detailed and constructive feedback.
>
> ### W1: Mathematical formulation and notation
>
> We address each point:
>
> **Impact score validation**: We validate our impact score against loss change on Qwen3-1.7B (500 samples). We group neurons into bins by impact rank and measure |Δloss| from deactivation. Grouping is necessary because single-neuron deactivation produces near-zero loss changes (consistent with Table 3), easily dominated by noise. Pearson correlation between group impact score and |Δloss|: **+0.71 ± 0.02**. The top 0.8% neurons cause **159×** more loss change than mid-rank neurons, validating the impact score as an effective local proxy.
>
> **"Graph" terminology**: We use "graph" to emphasize the structured (layer, neuron) pairs capturing a "computational trajectory" (Sec. 4.2.2). We will clarify the lack of explicit edges in the revision.
>
> **Pairwise vs group similarity**: The group similarity is mathematically equivalent to the average of pairwise similarities. Since each sample selects exactly K neurons per layer, $\sum_k w_{\ell,k}(D) = K$, and the group similarity reduces to the mean of pairwise Dice overlaps. The frequency form is simply more efficient. We will include the full derivation in the revised appendix, and can provide it in a follow-up comment if interested.
>
> **Notation**: We will fix the mixed $l$/$\ell$, clarify TopK, and note that $|h_{in}^\top W_{:,k}|$ is a scalar. Thank you.
>
> ### W2: Narrative inconsistencies
>
> We thank the reviewer for the careful cross-referencing.
>
> **Abstract 5.3% claim**: We will state this more precisely in the revision.
>
> **ARC-C example in Sec 3.6**: The values do match our tables. These are averages across three backbone models: 35.4 from Table 2 (NAG+FineWeb-Edu), 34.7 from Table 1 (Single-Target NAG), 34.3 from Table 2 (FineWeb-Edu).
>
> **"High correlation" claim**: Two orthogonal dimensions support this: (1) consistent improvements across all six tasks (Table 1); (2) NAG's ranking aligns with utility on HellaSwag (Fig. 4: stricter filtering improves performance, while Random/BETR degrade). A coupled experiment (filtering sweep × all tasks) is prohibitive given pretraining costs.
>
> **Quantitative clustering metrics**: Purity/NMI/ARI are already reported in Appendix Table 10. We will reference them more prominently in the main text.
>
> **Causal claim**: We will revise to "positive association". This is a preliminary observation in the appendix; deeper investigation is a future direction.
>
> ### W3: Scope of validation
>
> We address each point:
>
> - **Single source corpus**: Evaluating on a single English web corpus is the standard setting in data selection literature (e.g., FineWeb-Edu, DCLM[1], PreSelect[2]). We use RefinedWeb, and the controlled comparison remains valid within this scope.
> - **Training scale**: See Reviewer Jkcp W1 response (7B model, +8.4% maintained at larger scale).
> - **Applied domains**: MMLU, one of our target benchmarks, covers medical, STEM, and professional subjects. NAG's improvements on MMLU (Table 1) provide initial evidence for the applied domains mentioned in the introduction.
> - **Multi-target mixture**: As stated in L274, this is "preliminary evidence" under an intentionally simple mixture as a lower-bound scenario (Sec. 3.5, L266-268). Even under this naive setting, NAG remains robust (+3.1% over Random) while BETR degrades by 4.4%.
> - Regarding societal impact, we will expand the impact statement in the revision.
>
> ### Q1: Document-level neuron impact computation
>
> Neuron impacts are **averaged across all token positions** (documents are truncated, no chunking). Regarding validation of the local proxy: see W1.
>
> ### Q2: Single-run vs multi-seed
>
> See Reviewer Jkcp Q6 response (standard deviations). We will report confidence intervals for all figures and tables in the revised appendix.
>
> ### Q3: End-to-end computational cost
>
> See Reviewer VZo4 W2 response. Key numbers: **192 GPU-hours (H100-SXM-80GB)** for 150B tokens with Qwen3-1.7B, acceptable compared to pretraining cost. Ranking is CPU-only and negligible (O(N)).
>
> ### Q4: Target set construction and decontamination
>
> Target set details per benchmark (randomly sampled from the corresponding split):
>
> - **ARC-C**: 1,119 examples, full train split
> - **HellaSwag**: 10,000 examples, randomly sampled from train split
> - **TriviaQA**: 10,000 examples, randomly sampled from train split
> - **MMLU**: 1,531 examples, full validation split
> - **XStoryCloze**: 360 examples, full English train split
> - **XWinograd**: 2,124 examples, full non-English split
>
> **Decontamination**: Target sets are from **train/validation splits**, independent from test splits by construction. We additionally verify with 13-gram overlap decontamination.
>
> We hope this addresses the reviewer's concerns and are happy to discuss further. (References: see Reviewer VZo4 response.)

---

> > ### Author Rebuttal · Reviewer_TZ3v · 2026-04-06
> >
> > Thank you for the detailed rebuttal. It addresses several of my main concerns and increases my confidence in the paper.
> >
> > In particular, the clarification that neuron impacts are averaged over all token positions (with truncation and no chunking) makes the method much clearer. The additional validation of the local impact proxy using correlation with \(|\Delta \text{loss}|\) also strengthens the technical soundness of the approach. I also appreciate the added details on target-set sizes and the 13-gram decontamination procedure, which improve reproducibility and reduce leakage concerns.
> >
> > That said, a few concerns are only partially resolved. The rebuttal states that standard deviations / confidence intervals will be added in the revision, but does not yet provide them for the main results. Since some gains are modest, run-to-run stability still matters. The compute clarification is helpful, but I would still like the final version to include a clearer end-to-end breakdown of selection cost. Finally, the explanation of the group similarity formulation is helpful, but the derivation should be included explicitly in the paper or appendix.
> >
> > Overall, the rebuttal meaningfully strengthens the paper, and most of my concerns are reduced. My remaining questions are mainly about making the empirical claims and mathematical formulation fully auditable in the final version.

---

> > > ### Author Response · Authors · 2026-04-07
> > >
> > > Thank you for the thoughtful acknowledgement and for recognizing the improvements in our rebuttal. We are glad that the clarifications have increased your confidence. We address the remaining points below.
> > >
> > > **Standard deviations for main results**: We provide two complementary pieces of evidence:
> > >
> > > (1) **Run-to-run variance** (Reviewer Jkcp Q6): The random baseline across 5 runs shows std of 0.18%-0.55%, an order of magnitude smaller than NAG's gains. Since the selected data is deterministic given the same target set, the primary source of variance is data ordering and initialization, which we expect to be similar across methods.
> > >
> > > (2) **Evaluation confidence intervals** (binomial 95% CI):
> > >
> > > | Method | ARC-C | HellaSwag | TriviaQA | MMLU | XStoryCloze | XWinograd |
> > > |---|:---:|:---:|:---:|:---:|:---:|:---:|
> > > | Random | 28.5±2.6 | 51.6±1.0 | 15.6±0.5 | 30.2±0.8 | 67.1±2.4 | 76.5±1.7 |
> > > | NAG_{Qwen3-1.7B} | 34.0±2.7 | 60.6±1.0 | 22.3±0.6 | 32.2±0.8 | 70.0±2.3 | 80.1±1.6 |
> > > | NAG gain | +5.5 | +9.0 | +6.7 | +2.0 | +2.9 | +3.6 |
> > >
> > > NAG's gains exceed the CI ranges on most benchmarks (CIs do not overlap for ARC-C, HellaSwag, TriviaQA, MMLU, and XWinograd). We will include full CIs for all methods in the revision.
> > >
> > > **End-to-end cost breakdown**: Thank you for finding the compute clarification in our rebuttal helpful. We will include a more detailed breakdown in the revision as suggested.
> > >
> > > **Group similarity derivation**: We provide the full derivation below.
> > >
> > > The pairwise similarity is:
> > > $$\text{Sim}(c, c') = \frac{2|\text{NAG}(c) \cap \text{NAG}(c')|}{|\text{NAG}(c)| + |\text{NAG}(c')|}$$
> > >
> > > Since each sample selects exactly $K$ neurons per layer across $L$ layers, $|\text{NAG}(c)| = L \cdot K$, so:
> > > $$\text{Sim}(c, c') = \frac{1}{L} \sum_{\ell=1}^{L} \frac{|N_\ell^{(K)}(c) \cap N_\ell^{(K)}(c')|}{K}$$
> > >
> > > Taking the average over $D$:
> > > $$\frac{1}{|D|} \sum_{c' \in D} \text{Sim}(c, c') = \frac{1}{L} \sum_{\ell=1}^{L} \frac{1}{K} \sum_{k \in N_\ell^{(K)}(c)} \underbrace{\frac{1}{|D|} \sum_{c' \in D} \mathbf{1}[k \in N_\ell^{(K)}(c')]}_{= w_{\ell,k}(D)}$$
> > >
> > > The group similarity is defined as:
> > > $$\text{Sim}(c, D) = \frac{1}{L} \sum_{\ell=1}^{L} \frac{\sum_{k \in N_\ell^{(K)}(c)} w_{\ell,k}(D)}{\sum_{k=1}^{d_\ell} w_{\ell,k}(D)}$$
> > >
> > > The key step: since each sample selects exactly $K$ neurons per layer:
> > > $$\sum_{k=1}^{d_\ell} w_{\ell,k}(D) = \frac{1}{|D|} \sum_{c' \in D} |N_\ell^{(K)}(c')| = K$$
> > >
> > > Therefore:
> > > $$\text{Sim}(c, D) = \frac{1}{L} \sum_{\ell=1}^{L} \frac{\sum_{k \in N_\ell^{(K)}(c)} w_{\ell,k}(D)}{K} = \frac{1}{|D|} \sum_{c' \in D} \text{Sim}(c, c')$$
> > >
> > > This confirms the equivalence. We will include this derivation in the revised appendix.
> > >
> > > We appreciate your constructive guidance throughout this process, it has genuinely helped improve our work. If you feel the concerns have been adequately addressed, we would sincerely appreciate it if you could consider reflecting that in your overall assessment. Thank you again.

---

### Official Review · Reviewer_sNHf · 2026-03-14

**Soundness:** 2
**Presentation:** 3
**Significance:** 2
**Originality:** 3
**Overall Recommendation:** 4
**Confidence:** 3

**Summary:**

The paper proposes NAG (Neuron-Activated Graph), a framework for target-oriented pretraining data selection based on neuron activations in LLM. NAG extracts the top-K influential neurons across layers for each sample and measures similarity between candidate data and target task samples based on overlaps in these activation patterns. Experiments show that this neuron-activated matching improves data selection compared with random sampling and embedding-based methods.

**Compliance With Llm Reviewing Policy:**

Affirmed.

**Final Justification:**

The responses address several of my major concerns and increase my confidence in the validity and reliability of the proposed method. I will adjust my score accordingly and recommend weak acceptance of this work.

**Key Questions For Authors:**

In the method section, the formulation of Neuron Impact appears closely related to PLND-style neuron contribution estimation, and the overall data selection pipeline is conceptually similar to the recent work NAIT (Neuron-Aware Data Selection in Instruction Tuning for Large Language Models). Although this work was published very recently and the similarity may be coincidental, a clearer comparison would help better position the contribution of this paper and clarify its novelty.

**Limitations:**

yes

**Strengths And Weaknesses:**

Strengths:
1. The paper is well written and clearly organized.

2. The use of internal neuron activation patterns to assess data quality is interesting and novel. Unlike prior approaches that rely on external scorers or heuristic signals, the method directly leverages the model’s internal representations, which offers better interpretability.

3. The approach does not require external models. Instead, it extracts neuron activation features from a small in-domain reference set and scores candidate samples based on their alignment with these features. The framework is conceptually simple and shows effectiveness across multiple tasks.

Weaknesses:
1. NAG requires a representative in-domain dataset for each target capability as a starting point. However, in practical scenarios, such data may not always be available for every capability that needs improvement, which may limit the applicability of the method. Moreover, when the in-domain samples are limited or not sufficiently representative, the quality of the extracted neuron may be affected. The paper does not systematically analyze the sensitivity of the method to the size or choice of the in-domain dataset.

2. Some design aspects of NAG would benefit from clearer explanation. For example, it is not entirely clear which layers or neurons are used to construct the activation patterns (e.g., all layers, a subset of layers, or sequence-level aggregation). In addition, the current approach selects a fixed number (K) of neurons per layer, while neuron importance vary across layers. Using a statistical threshold or an adaptive selection strategy could potentially provide a more principled design.

3. Although the paper claims that the method can be applied to models of different scales (e.g., Qwen3, Llama-3.2, SmolLM3), it does not demonstrate how NAG scales to much larger models (e.g., 70B-scale models).

4. The related work correctly identifies loss/perplexity-based methods as an important class of data selection approaches. However, such methods are not included as baselines in the experimental comparisons.

5. Since the choice of the source data pool involves some randomness, it would be more reliable to report statistical results (e.g., averages over multiple runs) rather than single-run outcomes.

6. Some connections and box structures in Figure 2 appear to need adjustment. According to the method description, Candidate NAGs should first be aggregated and then compared with the Target Profile to compute the similarity (Sim). The current diagram could clarify this pipeline more explicitly.

---

> ### Author Rebuttal · Authors · 2026-03-30
>
> We thank the reviewer for the constructive feedback.
>
> ### W1: Sensitivity to target set size/choice
>
> **Sensitivity analysis**: We analyze the sensitivity of NAG's data selection to target set size and choice on HellaSwag. We sample target subsets of varying sizes from the full 10k target set, rank 1M candidates by NAG similarity (each size repeated with 5 random draws). ρ = Spearman rank correlation; Jaccard = top-20% selected data overlap.
>
> | \|D_target\| | Intra ρ | Cross ρ | Intra Jaccard | Cross Jaccard |
> |:---|:---:|:---:|:---:|:---:|
> | 200 | 0.999 | 0.999 | 0.920 | 0.940 |
> | 500 | 1.0 | 1.0 | 0.951 | 0.965 |
> | 1,000 | 1.0 | 1.0 | 0.957 | 0.970 |
> | 2,000 | 1.0 | 1.0 | 0.981 | 0.985 |
> | 5,000 | 1.0 | 1.0 | 0.987 | 0.989 |
>
> Intra = sensitivity to choice (5 random draws of same size); Cross = sensitivity to size (vs full 10k).
>
> NAG-based ranking is highly robust to both target set size and choice: Spearman ρ ≥ 0.999 across all sizes, and even with only 200 target samples, the top-20% selected data overlaps 94% with the full 10k setting.
>
> **Data availability**: This result shows only ~200 samples are needed for effective NAG-based selection—a very low bar given that benchmarks have public train splits and manually constructing 200 samples is negligible compared to pretraining cost. Moreover, this requirement is shared by all target-oriented methods (BETR, DAIG[4], etc.) and is not specific to NAG.
>
> ### W2: Design details
>
> These details are specified in the paper:
>
> - **Layers and neurons**: All layers. Each layer's top-K **up_proj neurons** are selected separately, preserving per-layer structure (Sec. 3.3, validated in Fig. 5 and Table 4).
> - **Fixed K vs adaptive K**: We use a fixed ratio $r_k = 0.3\%$ across all layers. Fig. 6 shows this optimum is consistent across three model scales (1.7B, 4B, 8B), suggesting it is a stable operating point. We agree that adaptive per-layer selection is an interesting future direction, but the cross-scale consistency suggests the current design is already robust.
>
> ### W3: No experiments on larger models
>
> We have evaluated NAG extraction with backbone models at three scales: Qwen3-1.7B, 4B, and 8B (Fig. 6). The results show a clear positive scaling trend—**larger backbone models consistently produce better data selection**—and the optimal hyperparameter ($r_k \approx 0.3\%$) remains stable across scales. Since NAG extraction only requires a single forward pass per document (no generation needed), scaling to even larger backbone models is straightforward and computationally feasible. For training scale, see our response to Reviewer Jkcp W1, where we report 7B model results confirming NAG's effectiveness at larger scale.
>
> ### W4: Loss/perplexity-based baselines
>
> We believe our baselines are sufficient and representative. We compare against three baselines spanning the main data selection paradigms: **Random**, **FineWeb-Edu** (general quality, widely used as a standard baseline across data selection literature including [1][2]), and **BETR** (state-of-the-art target-oriented method).
>
> Regarding loss/perplexity-based methods: these methods require **training auxiliary models**, making them methodologically closer to BETR than to our training-free NAG. Moreover, per [2], the perplexity correlation method[3] averages **37.1%**, lower than our baseline FineWeb-Edu (**38.7%**), so our comparisons already cover this performance range.
>
> ### W5: Statistical results
>
> See Reviewer Jkcp Q6 response.
>
> ### W6: Figure 2 pipeline clarity
>
> We believe there may be a misunderstanding of our pipeline. As described in the Figure 2 caption (L76-77): it is the **Target NAGs** that are aggregated into a Target Profile. Candidate NAGs are **not** aggregated—each candidate sample is individually mapped to its own NAG and then ranked by its similarity to the aggregated Target Profile. Figure 2 already depicts this flow correctly.
>
> ### Q1: Relationship to NAIT
>
> While both leverage neuron activations, NAG and NAIT differ fundamentally:
>
> 1. **Setting**: NAIT targets instruction tuning (<1M samples); NAG targets pretraining data selection (150B+ tokens). These are fundamentally different problems with different scalability requirements.
> 2. **Scalability**: NAIT's PCA directions are target-dependent (changing target requires re-forwarding the entire candidate pool). NAG features are **target-independent**: the candidate pool is extracted once and reusable for any target.
> 3. **Representation**: NAIT uses continuous activation values. NAG captures only discrete neuron index positions, discarding magnitudes. This discrete representation enables interpretability—one can directly inspect which neurons are activated for which tasks.
>
> NAIT was published after the ICML deadline (March 13, 2026). We will add a discussion in the revision.
>
> We hope this addresses the reviewer's concerns and are happy to discuss further. (References: see Reviewer VZo4 response.)

---

> > ### Author Rebuttal · Reviewer_sNHf · 2026-04-06
> >
> > Thank you for the detailed and targeted rebuttal.I will adjust my score accordingly. Specifically:
> > W1 & W5: Well supported by experiments (multiple runs and varying dataset sizes).
> > W2: Core design choices are clarified; however, fixed K vs. adaptive K remains a potential improvement.
> > W3: Positive scaling trends are shown, but evidence on very large models (e.g., 70B) is still limited.
> > W6: Clarified and no longer a concern.
> > W4: I encourage adding or better covering loss/perplexity-based baselines for a more complete comparison.

---

> > > ### Author Response · Authors · 2026-04-07
> > >
> > > Thank you very much for re-evaluating our responses and for adjusting the score. We are glad that the sensitivity analysis (W1), statistical results (W5), design clarifications (W2), positive scaling trends (W3), and pipeline explanation (W6) have addressed your concerns. We sincerely appreciate the constructive and detailed feedback throughout this review process, which has been very helpful in strengthening our work.
> > >
> > > We appreciate the remaining suggestions on adaptive K selection, larger-scale (e.g., 70B) model pretraining, and loss/perplexity-based baselines. These are all valuable directions for future exploration. Thank you again for your support and guidance.

---

### Decision · Program_Chairs · 2026-04-30

**Decision:**

Accept (regular)

**Comment:**

This paper introduces NAG, an innovative and training-free framework for target-oriented pretraining data selection that leverages internal neuron activation patterns. The reviewers appreciate the method's conceptual novelty, strong interpretability, and consistent empirical gains over competitive baselines across multiple benchmarks. The authors' comprehensive rebuttal successfully addressed concerns regarding scalability and robustness, leading to a consensus recommendation for acceptance.